



# Water Vapor Measurements inside clouds and storms using a Differential Absorption Radar

Luis Millán[1], Matthew Lebsock[1], Ken Cooper[1], Jose Siles[1], Robert Dengler[1], Raquel Rodriguez Monje[1], Amin Nehrir[2], Rory Barton-Grimley[2], James Collins[3], Claire Robinson[4], Kenneth Thornhill[4], and Holger Vömel[5]

[1]Jet Propulsion Laboratory, California Institute of Technology, Pasadena, CA, USA
[2]NASA Langley Research Center, Hampton VA, USA
[3]Coherent Application, Inc.- Psionic LLC, NASA Langley Research Center, Hampton VA, USA
[4]Analytical Mechanics Associates - NASA Langley Research Center, Hampton VA, USA
[5]National Center for Atmospheric Research, Boulder, CO 80301, USA

**Correspondence:** L. Millán (lmillan@jpl.nasa.gov)

**Abstract.** NASA's Vapor In-cloud Profiling Radar (VIPR) is a tunable G-band radar designed for in-cloud and precipitation humidity remote sensing. VIPR estimates humidity using the differential absorption radar (DAR) technique. This technique exploits the difference between atmospheric attenuation at different frequencies ("on" and "off" an absorption line) and combines it with the ranging capabilities of the radar to estimate the absorbing gas concentration along the radar path.

We analyze the VIPR humidity measurements during two NASA field campaigns: (1) the Investigation of Microphysics and Precipitation for Atlantic Coast-Threatening Snowstorms (IMPACTS) campaign, with the objective of studying wintertime snowstorms focusing on East Coast cyclones; and (2) the Synergies Of Active optical and Active microwave Remote Sensing Experiment (SOA$^2$RSE) campaign which studied the synergy between DAR (VIPR) and differential absorption lidar (DIAL, the High altitude Lidar Observatory - HALO) measurements. We discuss a comparison with dropsondes launched during these campaigns as well as an intercomparison against the ERA5 reanalysis fields. Thus, this study serves as an additional evaluation of ERA5 lower tropospheric humidity fields. In addition, we show a smooth transition in water vapor profiles between the in-cloud and clear-sky measurements obtained from VIPR and HALO respectively, which highlights the complementary nature of these two measurement techniques for future airborne and space-based missions.

*Copyright statement.* ©2023. California Institute of Technology. Government sponsorship acknowledged.

## 1 Introduction

Accurate measurement and understanding of tropospheric water vapor is crucial for improving our knowledge of cloud and precipitation microphysical processes, atmospheric radiative transfer, land-atmosphere interactions, and weather forecasting. Due to its importance, various methods have been developed and employed to measure water vapor from the ground, aircraft, and space. To date the water vapor profile of the lower troposphere is not well-observed from satellites (e.g., Teixeira et al.,



2021) and only well-characterized from a few surface remote sensing super sites (e.g., Wang et al., 2000). Thus, the 2017 Earth Science Decadal Survey (National Academies of Sciences and Medicine, 2018) recommended the development of new technologies to enhance the measurement of atmospheric thermodynamics, particularly within the planetary boundary layer, the lowest level of the troposphere. Active water vapor sounding techniques such as differential absorption lidar (DIAL) and differential absorption radar (DAR), have been proposed as potential solutions to obtain accurate high vertical resolution water
vapor profiles in clear sky and cloudy regions using DIAL and DAR respectively (Teixeira et al., 2021). These techniques exploit the difference between the backscatter signals (either a laser or a radar pulse) at different frequencies ("on" and "off" an absorption line) to estimate the gaseous absorption between the instrument and the scattering target. Previous studies have demonstrated the efficacy of DIAL and DAR in estimating water vapor profiles (e.g., Nehrir et al., 2017; Roy et al., 2022).

In this study, we present an analysis of airborne water vapor estimates obtained with the Vapor In-Cloud Profiling Radar
(VIPR) during two field campaigns. The first is the year 2 deployment of the NASA Investigation of Microphysics and Precipitation for Atlantic Coast-Threatening Snowstorms (IMPACTS) field campaign (McMurdie et al., 2022). The second is the NASA Synergies Of Active optical and Active microwave Remote Sensing Experiment (SOA$^2$RSE) campaign, which aimed to explore the synergy of DIAL and DAR measurements. While DIAL signals are sensitive to backscatter from both aerosols and molecules and attenuate rapidly in cloudy scenes, the much-longer-wavelength DAR signals are only sensitive to the larger par-
ticles in cloudy and precipitating scenes. Here we present the first ever demonstration of complementarity of these two active water vapor profiling techniques. We validate VIPR measurements of water vapor against colocated dropsonde measurements and reanalysis fields.

## 2  VIPR and the IMPACTS and SOA$^2$RSE campaign

VIPR is a G-band differential absorption radar (DAR) that was developed at NASA's Jet Propulsion Laboratory as a proof-
of-concept instrument. This all-solid-state radar operates on the flank of the 183 GHz water vapor line, in a band where radar operation can be made with permission of transmission regulation authorities, and utilizes frequency-modulated continuous-wave (FMCW) mode to significantly increase detection sensitivity compared to pulsed systems. Earlier iterations of the VIPR system have been deployed in different scenarios: on a ground-based platform to demonstrate its ability to accurately profile water vapor in cloud and precipitating scenes (Roy et al., 2018); as part of a multi-frequency radar deployment, to investigate
the sensitivity of a G-band radar to cloud liquid and ice microphysics when combined with lower frequency radars (Lamer et al., 2021); and, onboard a DHC-6-300 Twin Otter aircraft, to evaluate its performance from an airborne platform (Roy et al., 2021).

Subsequently, VIPR was modified to use two identical reflectors as separate primary apertures for the transmitting and receiving, in contrast to the previous configuration where a single primary reflector was used. This modification was implemented
to facilitate the integration of VIPR into NASA's P-3 aircraft, which utilizes radomes to safeguard the instruments against environmental factors. More significantly, VIPR also underwent modifications to expand its frequency coverage to approximately 158.6 - 174.8 GHz, as compared to the previous configuration of 167 - 174.8,GHz. The broader bandwidth enabled the addition



**Table 1.** VIPR system parameters

| Parameter | Value |
| --- | --- |
| Frequencies | 158.6, 167.12, 174.74 GHz |
| Transmit power | 120-210 mW |
| 3 dB beam width | $0.3^o$ |
| Antenna gain | 55 dB |
| Beam polarization | Circular |
| Receiver noise figure | 8 dB |
| Pulse duration | 1 ms |
| Number of pulses per frequency | $492^a$ |
| Radar duty cycle | 80% |
| Chirp bandwidth | 10 MHz |
| Native range resolution | 15 m |

[a] 246 chirp-up pulses and 246 chirp-down pulses

[b] assuming a 607 kmh$^{-1}$ average P-3 speed

of a third radar frequency further away from the water vapor line center which was implemented to mitigate retrieval biases associated with frequency-dependent hydrometeor extinction and backscatter (Roy et al., 2021). Table 1 summarizes the major

radar system parameters used in this work. Since the Twin Otter airborne measurements, VIPR was reconfigured to operate over a wider bandwidth, and to use two separate 38 cm diameter reflector antennas (one for transmit and one for receiver), instead of the previous single 60 cm diameter antenna shared between transmit and receive using a quasioptical duplexing subsystem. The wider tuning bandwidth allows three frequencies to be transmitted instead of two, while the use of separate antennas is compatible with the use of radomes that are necessary for VIPR's flights on higher-flying aircraft. As a consequence of the

wider-tuning feature, VIPR's transmit power for the measurements shown here was lower than those made previously.

The IMPACTS field campaign, sponsored by NASA, was an Earth Ventures Suborbital (EVS) field campaign aimed at studying high impact winter snowstorms, particularly cyclones that affect the US East Coast (McMurdie et al., 2022). The primary observation platforms for the IMPACTS campaign were the ER-2 and P-3 aircraft. The second deployment of the IMPACTS campaign was conducted from NASA Wallops during January and February 2022. As part of the P-3 payload,

VIPR was deployed during four flights, as outlined in Table 2 and depicted in Figure 1-top. Most of the VIPR measurements were obtained during flights along the East Coast northward of NASA Wallops, with the exception of February 17th flight, which targeted an in-land snowstorm system.

Following the IMPACTS field campaign, VIPR conducted four additional flights to assess its synergy with the High Altitude Lidar Observatory (HALO) instrument (Carroll et al., 2022) as part of the SOA$^2$RSE campaign, which was sponsored by

NASA's Earth Science Technology Office (ESTO). The purpose of SOA$^2$RSE was to demonstrate the synergies of the DIAL and DAR measurement approaches for the first time. These flights were aimed towards a range of clear sky and cloudy condi-



**Table 2.** VIPR IMPACTS and SOA$^2$RSE flights details and objectives

|  | Flightdate | VIPR turn-on | Duration | Objectives |
|---|---|---|---|---|
|  |  | UTC | hours |  |
| IMPACTS | 2022-01-14 | 19:38 | 5.3 | To target precipitation bands offshore of Cape Cod and in the Gulf of Maine |
|  | 2022-02-13 | 11:48 | 3.3 | To target a frontal boundary across the mid-Atlantic region (mid-level frontogenesis was the primary forcing for clouds and precipitation) |
|  | 2022-02-17 | 16:26 | 8.1 | To sample a developing storm over the Midwest (over southern Lake Michigan and Indiana) |
|  | 2022-02-25 | 15:28 | 3.6 | To target a warm-frontal system in eastern New York (snow and mixed-precipitating bands) |
| SOA$^2$RSE | 2022-03-04 | 13:22 | 7.3 | To explore the synergy of DIAL and DAR, targeting a frontal system warm and cold sector |
|  | 2022-03-07 | 15:50 | 4.9 | To explore the synergy of DIAL and DAR, targeting stratocumulus clouds and fair weather conditions for VIPR calibration |
|  | 2022-03-08 | 12:58 | 7.3 | To explore the synergy of DIAL and DAR, targeting multiple transects of convective system |
|  | 2022-03-09 | 15:30 | 3.8 | To explore the synergy of DIAL and DAR, targeting multiple transects of convective system |

tions over the Western Atlantic ocean near NASA Wallops Flight Facility. See Table 2 and Figure 1-bottom for details. Overall, these flights (IMPACTS and SOA$^2$RSE) encompass around 44 hours of VIPR data.

## 2.1 VIPR measurements and noise subtraction

A full description of the VIPR implementation is provided elsewhere (Cooper et al., 2018; Roy et al., 2020; Cooper et al., 2021; Roy et al., 2022). In short, VIPR is a Frequency Modulated Continuous Wave (FMCW, see Figure 2) radar measuring radar reflectivity profiles inside clouds and precipitation at three frequencies (158.6, 167.12, and 174.74 GHz). The fundamental measurement at each frequency, $f$, is a power spectrum as shown in Figure 3. These power spectra are converted to profiles of measured reflected power, $P_m(r, f)$, at each range bin, $r$, using the standard FMCW signal processing approach. (e.g., Cooper

et al., 2011). The measured power is the combination of the actual echo power, $P_e(r, f)$, and the noise power, $P_n(r, f)$, that is, $P_m(r, f) = P_e(r, f) + P_n(r, f)$.

The first step in our data analysis is to estimate $P_n(r, f)$ in each radar range bin so that we can then estimate $P_e(r, f)$ as $P_e(r, f) = P_m(r, f) - P_n(r, f)$. Noise in the VIPR profiles take two forms: (1) thermal noise, which is a characteristic of VIPR's receiver and the scene brightness temperature, and (2) phase noise that is carried by the radar's transmit signal, and





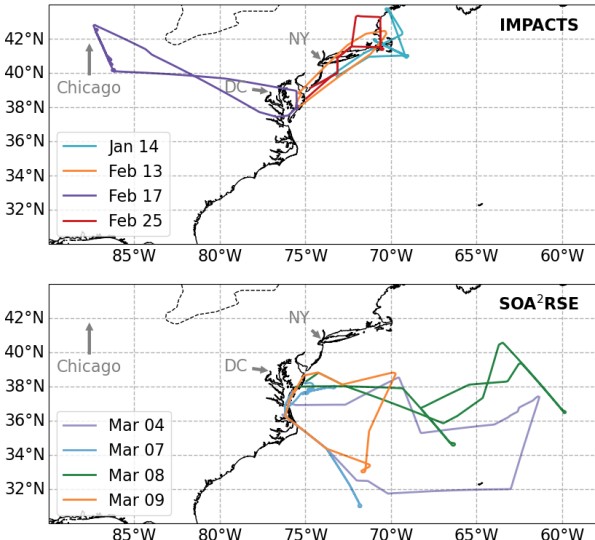

**Figure 1.** (top) Flight tracks of the P-3 VIPR measurements during the 2022 IMPACTS deployment. (bottom) Flight tracks of the P-3 VIPR and HALO measurements during the SOA$^2$RSE deployment. The maps display the East Coast of the US. The border with Canada is shown with a dashed line. Different flight-dates are color coded.

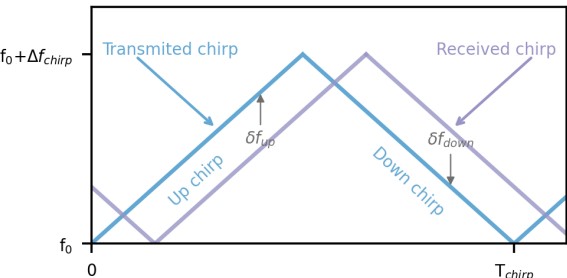

**Figure 2.** Schematic of FMCW radar ranges. The received chirp is offset in time from the transmit chirp, resulting in an instantaneous frequency difference for a point-target of $\delta f$, for most of the detection interval, that is proportional to the range of the target according to $\delta f = 2\Delta F_{chirp} r / c T_{chirp}$, where $\Delta F_{chirp}$ is the chirp bandwidth and $T_{chirp}$ is the chirp duration. Hence, the IF signal of the receiver mixer in an FMCW radar will have a frequency shift proportional to the range to target. For scenes with hydrometeors at multiple ranges, the IF will contain a linear superposition of frequencies representing the different ranges. Signal processing techniques are then used to convert the IF time-domain signal to a range-resolved power spectrum.

hence is more prominent with brighter reflectivity scatterers. The thermal noise can be modeled as $P_{nT} = kT_{sys}B$, where $K$ is Boltzman's constant, $T_{sys}$ is the system temperature and $B$ is the detection bandwidth. Phase noise originates in the radar's local oscillator used to generate the G-band signals, and it causes range side-lobes that spread throughout the observed power spectrum.





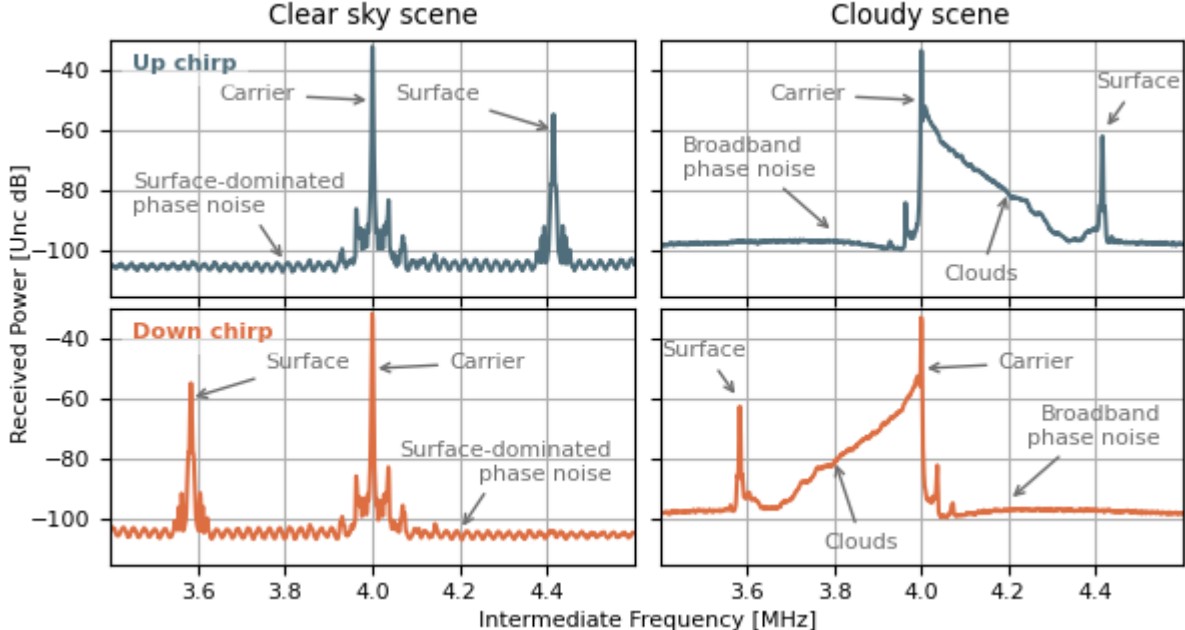

**Figure 3.** Power spectrum examples at 167.12 GHz for a clear sky and a cloudy scene. The surface return can be seen around 3.6 or 4.4 MHz depending on the chirp direction. Surface phase noise sidelobes are clearly seen in the clear sky case.

Through the IMPACTS and SOA²RSE flights $P_n(r, f)$ is dominated by phase noise, with two distinct forms: a surface-dominated phase noise, and broadband phase noise (resulting from multiple echos originating from clouds and precipitation that are distributed in range). See Cooper (2022) for more details. To illustrate this, Figure 4-top shows a curtain plot of the raw radar reflectivities at 167.12 GHZ during a 40 minute acquisition over the Atlantic Ocean on January 14, 2022. The surface can clearly be seen around 6 km range as a bright reflection practically at a constant range throughout this curtain. A periodic (in range) clutter signal associated with this reflection is visible above and below the surface. These range side lobes that appear in clear-sky profiles arose from surface-dominant phase noise. As the extent of the clouds and attenuation increases (towards the end of the acquisition), the intensity of the surface-reflection and the intensity of this periodic clutter also diminishes. In these profiles, the bright clouds produce their own broadband phase noise that overpowers the phase noise contribution from the surface reflection. That is, the phase noise power from the clouds is broadband, not lobed as for phase noise from a surface reflection, because the clouds extend continuously over a broad range so that there is no dominant lobed-noise contribution from a single range (Cooper, 2022).

To illustrate this further, Figure 4-right shows three slices of this curtain. In all three slices the transmit/receive leakage signal can clearly be seen at the aircraft altitude (i.e., at zero range). Additionally the surface return can be seen around 6 km. The first slice (S1, navy line) shows a clear sky scenario where the clutter induced by the surface return can clearly be seen above and below the surface; that is, the periodic signal with an amplitude of 1dB and a period of 300 m. The second slice (S2, light blue



line) portrays a partly cloudy scene with clouds close the aircraft and at a range of around 2.7 km. The periodic clutter, caused by the surface return, is still visible below and above the surface, but its amplitude has been reduced by the extra attenuation introduced by the clouds. The third slice (S3, green line) shows a cloud extending from the aircraft up to a ~4.8 km range. In this slice, the additional cloud attenuation has significantly attenuated the surface return leaving no discernable periodic phase noise from the surface reflection, but instead causing a broadband noise increase evident below the surface.

Two potential noise subtraction approaches can be applied to the VIPR reflectivities, one associated with the surface-dominant phase noise and the other due to broadband cloud induced phase noise. Both can mask reflections from significantly weaker targets at other ranges. To decide which type of noise subtraction to use, we first model the phase noise associated with the surface following the model described by Cooper (2022). In short, at a given separation in Intermediate Frequency (IF) between the carrier and the target location, $\Delta f_{IF}$, the point-target phase noise can be simulated as

$$\varphi(\Delta f_{IF}) = A + 10log_{10}(F) + B\Delta f_{IF} + C\exp\left(\frac{-\Delta f_{IF}^2}{D^2}\right) \tag{1}$$

where $A$ is 58 dBc (i.e., decibels relative to carrier), $B$ is -7 dB/MHz, $C$ is -3 dB, and $D$ is 0.2 Mhz. Lastly, $F$ is the interference modulator factor given by,

$$F = 4\sin^2\left(\frac{2\pi R}{c}\Delta f_{IF}\right) \tag{2}$$

where $R$ is the range to the target location related to the IF following $f_{IF} = 2KR/c$ where $K$ is the chirp-rate in Hzs$^{-1}$ and $c$
is the speed of light.

Once we have estimated the estimated surface phase noise, we perform a Fast Fourier Transform (FFT) on the signal below the surface to identify its frequency components. We then analyze the resulting frequency spectrum to see if a component matches the simulated periodicity of the modeled phase noise surface return (as computed by Equation 1). If such a match exists we estimate the range-dependent phase noise using the observed sub-surface measured power. We then use the fact that
the phase noise is symmetric about the surface to subtract the phase noise from the atmospheric ranges. If the FFT analysis does not match such component, we instead proceed to subtract the broadband phase noise derived from the opposite chirp direction as described by Roy et al. (2018).

Figure 5-top shows examples of this point-target phase noise model (light gray lines). As shown, on Slice 1 (a clear sky scenario), the modeled phase noise originating from the surface agrees with the observed noise below the surface, providing
confidence in the point-target noise model. In Slice 2 (a partly cloudy case) and Slice 3 (a cloudy case), the noise below the surface is well represented by the integration of the point-target models. These point-target curves exhibit a periodicity that is inversely proportional to the range of the carrier tone, along with an amplitude variation in the intermediate frequency. Note how the surface return point-target phase noise model reproduces the behaviour discussed in Figure 4, where the amplitude of its periodic clutter decreases as the surface return gets more attenuated. For instance, in Slice 3 the surface return point-target
phase noise model is orders of magnitude smaller than the surface return point-target phase noise model displayed in slice 1.

Figure 4-middle shows the same curtain as in the top panel after the noise subtraction. As expected, the contrast between the noise floor and the hydrometeor targets increases improving the dynamic range of VIPR, which improves the detection of





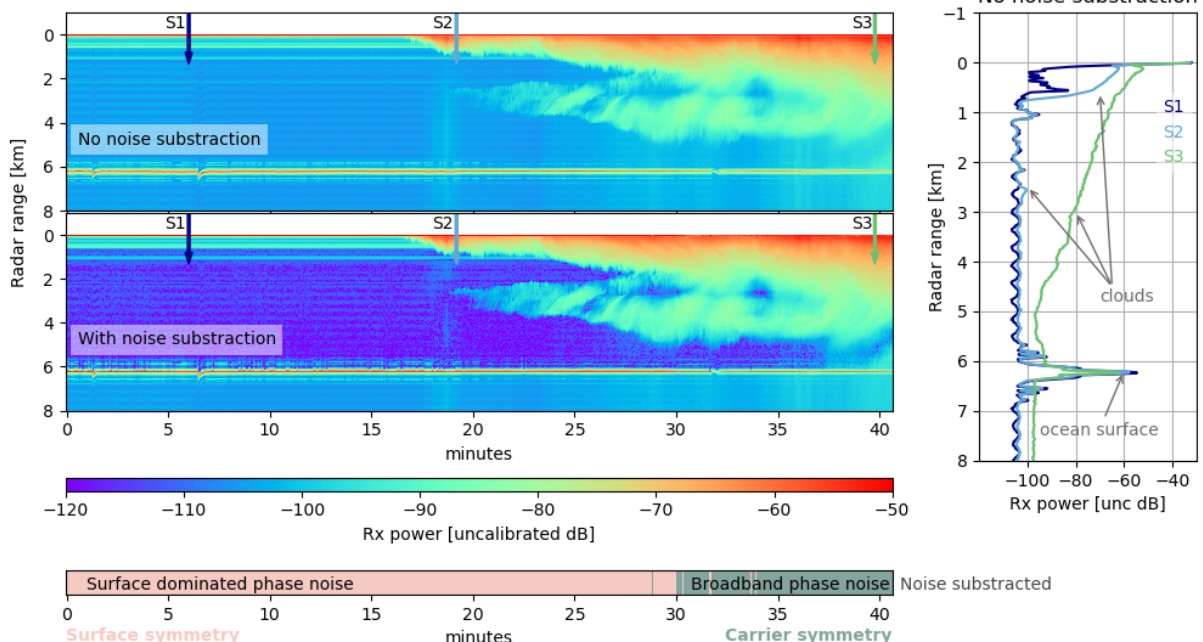

**Figure 4.** (top) Uncalibrated VIPR 167.12 GHz reflectivities measured during the first ~40 minutes of the January 14 flight. (middle) Uncalibrated VIPR 167.12 GHz reflectivities after the appropriate noise has been subtracted. (bottom) Noise subtracted (pink for phase noise and teal for thermal noise). (right) Three slices of the uncalibrated radar curtain (prior to the noise subtraction), showing situations with clear sky (S1, navy line, a bright reflection around 6.5 km), a partly cloud scene (S2, light blue), and a cloudy scene (S3, green).

weak clouds, such as in slice 2 around 2.7 km in range. As a reference, Figure 4-bottom displays which noise was subtracted through-out the flight.

## 2.2 Radar detection confidence flag

We create a hydrometeor confidence flag to identify ranges at which the observed reflectivity is associated with an atmospheric target as opposed to noise. This flag is based upon the phase noise model described in the previous section.

The simulated phase noise model as measured by VIPR, $\varphi_M$, is given by the sum of the point-target phase noise contributions, that is,

$$\varphi_M = \sum_{carrier}^{surface} \varphi(\Delta f_{IF}). \tag{3}$$

As shown in Figure 5 (dark gray lines), the summed phase noise agrees with the observed noise below the surface for the three slices. These slices covered different hydrometeor burdens: a clear sky scenario, a partly cloudy, and a cloudy case for slice 1, 2, and 3 respectively.



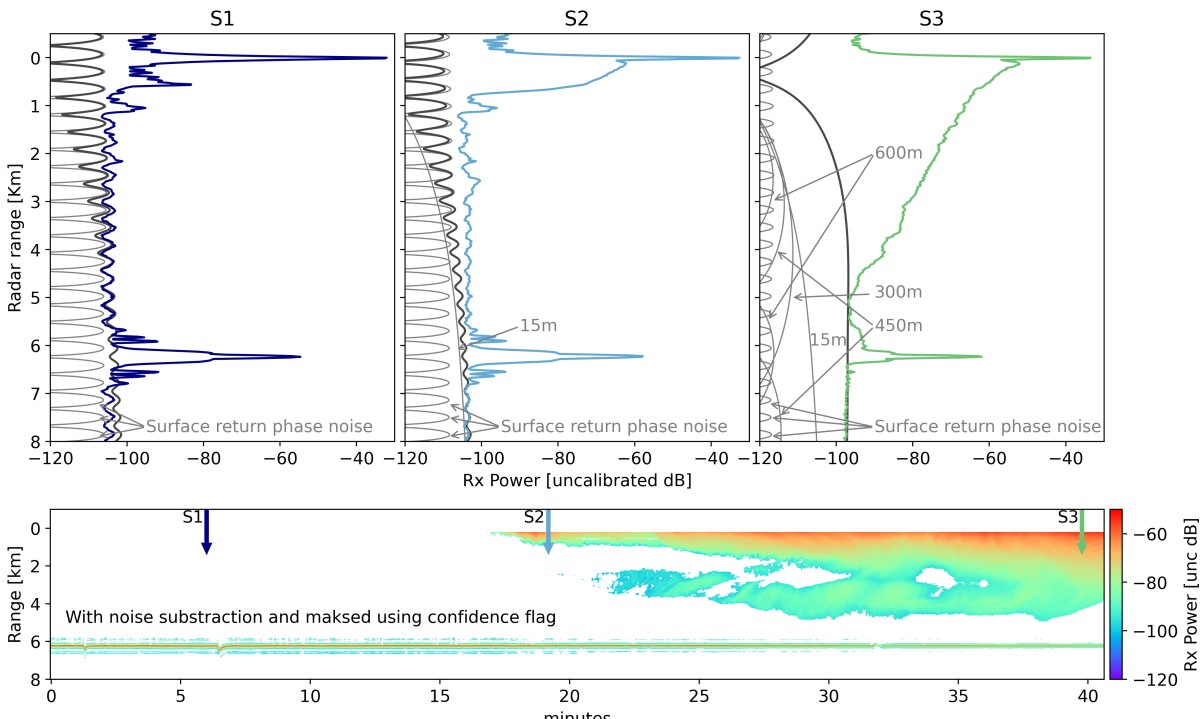

**Figure 5.** (top profiles) Radar reflectivities at the 3 slices discussed in Figure 4. The integrated phase noise model is shown in dark gray. Phase noise model examples at different ranges are shown in light gray. (bottom) Uncalibrated VIPR 167.12 GHz reflectivities measured during the first 40 minutes of the January 14 flight on (i.e., the same radar reflectivities shown in Figure 4) with noise subtraction as well as applying the confidence flag.

To be considered a radar detection, the returns need to be at least 3 dB greater than the envelope of the integrated phase noise model and to have a signal-to-noise ratio greater than 3. In those cases, the confidence flag is set to one (otherwise it is set to zero). In this context, the detection envelope refers to a curve following the local maxima of the phase noise model to avoid the local minima associated with the periodicity of the surface-reflection phase noise. This criteria allow us to discriminate spurious returns that at first sight could be consider clouds when in reality is just phase noise rising above the noise floor. See for example Figure 5-bottom in comparison with Figure4-bottom, by the end of the acquisition (after minute 37).

## 2.3 Calibrated Radar Reflectivity

After noise subtraction we calculate the calibrated radar reflectivity as

$$Z_e(r, f) = P_e(r, f)C(f)r^2 \qquad (4)$$



where $C(f)$ is a calibration factor, $Z_e(r,f)$ is the reflectivity of a cloudy volume in conventional meteorological units of decibels of mm$^6$/m$^3$. VIPR calibration was performed in a ground-based setting using a metallic spherical reflector suspended between two poles by a nylon thread. In simple terms, the calibration factor sets an absolute scale of echo power based on the well-known scattering cross section of the calibration sphere. Details of this procedure can be found in Cooper et al. (2021) and in Roy et al. (2021).

Errors sources in this calibration factor include: (1) inaccurate knowledge of the radar's parameters (such as potential drifts in transmit power and receiver noise between the ground calibration and airborne deployment), (2) uncertainties in the water vapor attenuation assumed for the beam path between the instrument and calibration sphere, (3) positioning of the calibration sphere with respect to the beam center, and (4) possible scattering effects of the suspending nylon thread. Based on previous calibration efforts this calibration method suggest an accuracy of 1 to 2 dB (Cooper et al., 2021; Roy et al., 2021), although it may be slightly larger as discussed in section 5.

The calibration we use was performed prior to aircraft installation on September 28, 2021, and also immediately following the removal of VIPR from the P-3 aircraft on March 15, 2022. The re-calibration was done because part way through the campaign, VIPR's low-noise amplifier failed and was replaced.

## 3 Retrieval methodology and datasets used for comparisons

### 3.1 Water vapor profiling retrieval

The DAR profiling technique (e.g., Roy et al., 2018; Battaglia and Kollias, 2019) begins by combining the observed reflectivities to form the observed absorption coefficient at two different ranges $r_1$ and $r_2 = r_1 + R$,

$$\gamma(r_1, r_2, f) = \frac{1}{2R} ln \left[ \frac{Z_e(r_1, f)}{Z_e(r_2, f)} \right] \tag{5}$$

where $Z_e(r, f)$ is the measured reflectivity (after noise subtraction) given by,

$$Z_e(r, f) = Z_{eff}(r, f) e^{-2\tau(r,f)} \tag{6}$$

$Z_{eff}(r, f)$ is the effective unattenuated reflectivity for a given target, and $\tau(r, f)$ is the one-way optical depth from the radar to the range $r$.

Thus, Equation 5 can be rewritten as,

$$\gamma(r_1, r_2, f) = \frac{1}{2R} ln \left[ \frac{Z_{eff}(r_1, f)}{Z_{eff}(r_2, f)} e^{-2\beta(r1,r2,f)R} \right] \tag{7}$$

where

$$\beta(r1, r2, f) = \frac{\tau(r_1, f) - \tau(r_2, f)}{R} \tag{8}$$

$$= \frac{1}{R} \int_{r_1}^{r_2} \left[ \sum \rho_j(r) \kappa_j(r, f) + \beta_p(r, f) \right] dr \tag{9}$$



is the average absorption coefficient between $r_1$ and $r_2$, $\rho_j(r)$ is the density of the $j$th gas, $\kappa_j(r, f)$ is the corresponding mass extinction cross section (which varies due to pressure and temperature variations along the radar path), and $\beta_p(r, f)$ is the particulate extinction coefficient. Note that the observed absorption coefficient, $\gamma(r_1, r_2, f)$, is not affected by absolute calibration, that is, the DAR profiling retrieval is self calibrated. Our calibration procedure described above is instead only relevant to the absolute-dBZ mapping of the clouds and precipitation using VIPR.

This equation can be simplified by separating the water vapour components from the other gases,

$$\beta(r_1, r_2, f) = \overline{\rho_v}(r_1, r_2)\kappa_v(f) + \overline{\beta_{dry}}(r_1, r_2, f) + \overline{\beta_p}(r_1, r_2, f) \tag{10}$$

where $\overline{\rho_v}$ is the water vapor density, $\kappa_v(f)$ is the water vapor mass extinction coefficient, $\overline{\beta_{dry}}(r_1, r_2, f)$ is the dry-air absorption coefficient and where the overline indicates taking the average between $r_1$ and $r_2$.

Thus equation 7 can be rewritten as,

$$\gamma(r_1, r_2, f) = \overline{\rho_v}(r_1, r_2)\kappa_v(f) + \frac{1}{2R}ln\left[\frac{Z_{eff}(r_1, f)}{Z_{eff}(r_2, f)}\right] \tag{11}$$

$$+ \overline{\beta_{dry}}(r_1, r_2, f) + \overline{\beta_p}(r_1, r_2, f) \tag{12}$$

which implies that the average humidity between $r_1$ and $r_2$ can be extracted by performing a least square fit to this equation, where the last 3 parameters can be assumed to vary weakly with frequency and contain information about the relative reflectivity of the two ranges in question, the dry air gaseous absorption, and the particulate extinction respectively. In principle, all 3 with a weak frequency dependence.

As shown by Roy et al. (2018), when using two radar tones at different frequencies, the humidity can be estimated directly using,

$$\overline{\rho_v}(r_1, r_2) = \frac{\gamma(r_1, r_2, f_2) - \gamma(r_1, r_2, f_1)}{\kappa_v(f_2) - \kappa_v(f_1)} \tag{13}$$

which assumes that the last 3 terms in equation 7 are a frequency independent offset. However this may not be necessarily true in regions where the particle size distributions of hydrometeors vary significantly in range, such as near cloud boundaries or in regions with phase changes. In those situations, the range-dependent differential scattering of hydrometers can be erroneously attributed to water vapor, resulting in measurement bias. If more than two frequencies are used, the retrieval can partly disentangle the differential extinction from the water vapor from the hydrometeor scattering and absorption effects as shown by Battaglia and Kollias (2019) and Roy et al. (2022).

The uncertainty in $\gamma(r_1, r_2, f)$ can be computed using standard error propagation techniques (Roy et al., 2018; Battaglia and Kollias, 2019). This uncertainty can then be used to estimate the humidity uncertainty. In this study, from the 15 m native VIPR reflectivities resolution, we use steps of 300 meters (i.e, R = 300 m) to derive the in-cloud profiles; which have an associated precision of around 5.5 g m$^{-3}$. This results in a water vapor profile with 300 m vertical resolution sampled every 15 m. These oversampled estimates are then averaged to a 300 meters vertical grid to smooth out the retrievals, improving the precision by a factor of $\sqrt{20}$ (i.e., to around 1.23 g m$^{-3}$). These type of water profiles are taken every 1.9 seconds, which is the time that it takes VIPR to cycle through the 3 frequencies for the programmed number of pulses per frequency. This temporal resolution results in an approximate along-track resolution of 320 m assuming a 607 km h$^{-1}$ average P-3 speed.



## 3.2 Partial column water vapor retrieval

220 As shown by Roy et al. (2022), when using two radar tones at different frequencies, the partial column water vapor (pCWV) can be estimated using,

$$pCWV = \frac{cos(\theta_i)}{2\langle \Delta k_v \rangle} \frac{\sigma_m^0(f_1)}{\sigma_m^0(f_2)} \tag{14}$$

where $\theta_i$ is the beam pointing angle relative to the local nadir, $\sigma_m^0(f)$ is the measured surface normalized radar cross section at a given frequency and

225 $$\langle \Delta k_v \rangle = \frac{\int_0^{R_s} \rho_v(r') \Delta k_v(r') dr'}{\int_0^{R_s} \rho_v(r') dr'} \tag{15}$$

and $\Delta k_v(r) = k_v(f2, T(r), p(r)) - k_v(f_1, T(r), p(r))$ is the local differential absorption cross section for water vapor per unit mass, that changes with temperature, $T(r)$, and pressure, $p(r)$ along the radar path. Thus by assuming a given temperature and pressure profile, and the shape of a water vapor profile, pCWV can be inferred. That is, the normalization of equation 15 ensures that the pCWV retrieval only depends on the humidity profile shape assumed.

230 As with the profile retrieval, when using only two frequencies the inferred pCWV may be subject to biases induced by frequency dependent attenuation contributions from the cloud and precipitation hydrometeors along the radar path. If more that two frequencies are used the retrieval can distinguish the differential extinction from the water vapor from the hydrometeor absorption effects as shown by Roy et al. (2020).

Note that for these retrievals accurate relative calibration factors are needed. The pCWV precision at the native VIPR 235 resolution is $\sim 0.5\,\mathrm{g\,cm^{-2}}$ which corresponds to the time that it takes VIPR to cycle trough the 3 frequencies, that is, $1.9\,\mathrm{s}$ (approximately $320\,\mathrm{m}$).

## 3.3 Ancillary Retrieval Information

Both the profiling and the pCWV retrievals require ancillary pressure and temperature information. This information is obtained from the Modern-Era Retrospective Analysis for Research and Applications version 2 (MERRA-2) reanalysis (Gelaro et al., 240 2017) which provides fields every $3\,\mathrm{h}$ on a 0.625 by $0.5\,^{\circ}$ longitude/latitude grid with a vertical resolution better than $1\,\mathrm{km}$ in the lower troposphere. In the retrievals, we simply find the closest grid point to the VIPR measurement in the $12\,\mathrm{UTC}$ MERRA-2 fields. Note that, as shown by Roy et al. (2018), the retrieved humidity is weakly dependent on the assumed pressure and only accrues a 10% error for an 8 K temperature deviation. For the pCWV retrievals the retrieval also assumes the normalized shape of the humidity profile (Roy et al., 2022), which is also derived from MERRA-2. This profile shape is needed to quantify the 245 absorption line broadening which depends on temperature and pressure. The assumed water vapor profile shape is then scaled by a multiplicative factor to match the observed ratios of the radar surface reflectivity. While we use MERRA-2 for ancillary information, we later use ERA5 as an independent reanalysis dataset against which the retrievals are compared as detailed in section 3.5.



All retrievals shown here only use radar returns classified as confident using the flag described in section 2.2. In this manuscript we only show results from the chirp-up estimates. We acknowledge that the chirp-down estimates are nearly identical. If we were to average both chirp-up and chirp-down measurements, the improvement in estimate precision would be a factor of $\sqrt{2}$.

### 3.4 Dropsondes

Dropsondes were deployed from the P-3 using the Advanced Vertical Atmospheric Profiling System (AVAPS) operated by NASA Langley Research Center (Hock and Franklin, 1999). The AVAPS Global positioning System (GPS) dropsondes measure pressure, temperature, humidity, wind speed, and wind direction as the parachuted dropsonde descends towards the surface. The AVAPS system has been used in many field campaigns (e.g. Wang et al., 2015; Sorooshian et al., 2019; McMurdie et al., 2022; Reid et al., 2023). The dropsondes used in the flights used in this study were the NCAR Research Dropsonde NRD41 manufactured at NCAR. Typical uncertainties are with $\pm 0.5$ hPa, $0.2$ C, $3\%$, and $0.5\,\mathrm{ms}^{-1}$ for pressure, temperature, humidity, and wind respectively.

The NRD41 and RD41 dropsonde family uses a heated humidity sensor, which has minimal biases measuring water vapor inside clouds and thus is well suited for comparisons of humidity measurements in cloudy environments. In addition, the direction of the profile during descent contains no significant risk for icing of the sensors. Prior to take-off of each research flight, the humidity sensors of all dropsondes were properly reconditioned to minimize any calibration drifts that may have happened between production and use of the dropsondes. No correction needed to be applied for the time response of the dropsonde humidity sensors during IMAPCTS and SOA$^2$RSE, because the temperature range for these observations was sufficiently warm and the humidity sensor sufficiently fast.

The dropsondes were processed through the Atmospheric Sounding Processing ENvironment (ASPEN, Martin and Suhr, 2023), to apply all required quality control and correction algorithms. Dropsonde data collected during IMPACTS and SOA$^2$RSE were not transmitted to the WMO Global Telecommunication System and were not assimilated in models. Therefore, the reanalysis comparison shown in the subsequent sections are not influenced by the dropsonde data and represent a truly independent analysis of the atmosphere in this region.

### 3.5 ERA5 reanalysis fields

We use data from the European Center for Medium-range Weather Forecasts (ECMWF) global reanalysis fifth generation (ERA5) hourly fields for intercomparison with the VIPR retrievals. ERA5 combines a high-resolution atmosphere modeling system with observations via a 4-D variational data assimilation. Extensive conventional and satellite observations of humidity are assimilated such as radiosondes as well as radiances from the Microwave Humidity Sounder and the Microwave Humidity Sounder 2 instruments on board NOAA-18 and -19, and FY-3C, along with radiances from the Meteorological Operational satellites. ERA5 provides an hourly record of the ocean, land, and atmospheric state with a $\sim 31$ km horizontal resolution and 137 levels covering from the surface to 0.01 hPa (Hersbach et al., 2020). To ease comparison, the ERA5 humidity fields were interpolated to the VIPR measurement times and locations.



To the best of our knowledge, limited comparisons have been made between ERA5 humidity fields and other datasets. A study by Gamage et al. (2020) compared ERA5 with unassimilated-sondes launched from Payerne, Switzerland. That comparison suggests that below ∼1.5 km ERA5 relative humidity is too dry (up to 20%RH), while between 1.5-5 km ERA5 is

8%RH wetter. Comparisons with COSMIC-2 water vapor suggest that these datasets agree overall within 6-12%, with larger difference in regions with frequent convection (Johnston et al., 2021). Thus, our study serves as an additional validation albeit limited for ERA5 lower tropospheric humidity fields.

## 4  Vapor Profile Results

Figure 6 shows the curtain plots of the radar reflectivities at 167.12 GHz through the IMPACTS and SOA$^2$RSE flights (after

applying the confidence flag and subtracting the appropriate noise). The effects of attenuation by a combination of liquid hydrometeors and water vapor are clearly seen in these reflectivities. For example, note the attenuation or disappearance of the surface return below regions of heavy hydrometers burden, such as on the February 17 flight around 420 minutes into the flight.

An overview of each flight is given below:

– January 14: Flight over a band of precipitation near the Gulf of Maine. The flight was largely over water therefore this was the best flight for VIPR retrieval validation with 11 sondes encompassing cloudy regions mostly from 2 to 6 km. Bands of relatively high reflectivity can be seen along the curtain for roughly 5 hours. Attenuation of the reflectivity profile below 2 km altitude is evident throughout much of the flight.

– February 13: Flight over a snow band along the coast extending between the Mid-Atlantic and Cape Cod. Three drop-
sondes were launch during this flight but only the first one over cloudy regions.

– February 17: Flight over a strong winter system that impacted the Great Lakes region, mostly over land so no dropsondes were launched during this flight. This flight offers an almost continuous curtain of radar reflectivites for eight hours. These were the largest reflectivities observed by VIPR during the deployment, which significantly attenuate the radar signal.

– February 25: Flight over a warm-frontal system in eastern New York. Note that the last four dropsondes are over clear sky so they are only useful for validating VIPR pCWV measurements.

– March 4: The first SOA$^2$RSE flight. This was a mostly a clear sky flight over ocean with some shallow convective clouds and light precipitation associated with a convergence line near the Bahamas mid-flight.

– March 7: A clear sky flight over ocean.

– March 8: An over ocean flight with a mix of clear sky and mid-level convection some of which heavily attenuates the radar.





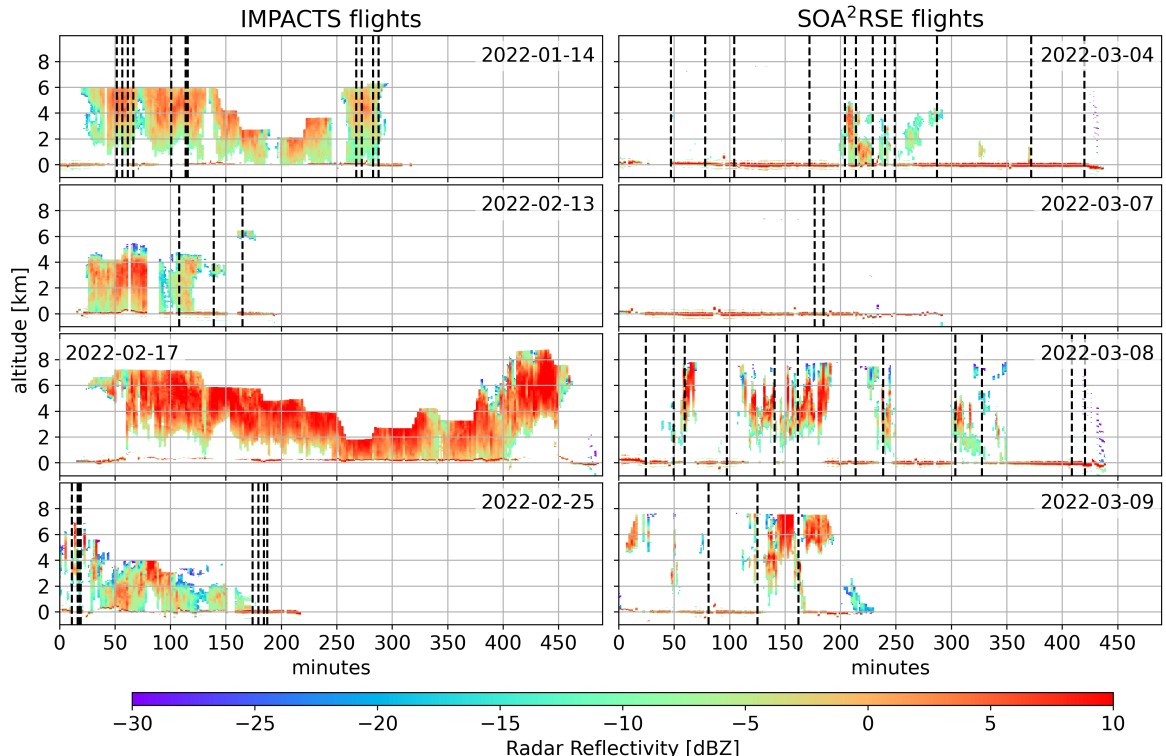

**Figure 6.** Calibrated radar reflectivities at 167.12 GHz for the IMPACTS (left) and the SOA²RSE (right) flights. Dropsondes launches are indicated by dashed vertical lines.

– March 9: An over ocean flight with a mostly clear sky, some deep convection and scatter shallow cumulus.

We now present results of the VIPR water vapor profiling retrievals. Figure 7 shows the curtain plots of the VIPR water vapor estimates for the IMPACTS flights and Figure 8 for the SOA²RSE flights. To reduce noise, these curtains show one minute averages of the VIPR retrievals, that is, up to 30 profiles that are taken every ∼2 seconds. Assuming a flight speed of 607 km/hr this corresponds to an along track averaging distance of ∼10 km. Note that the average is performed on the water vapor retrieval instead of the radar reflectivites because the mapping of radar reflectivity ratios into water vapor is non-linear. For intercomparison and to help with the interpretation, these figures also show the curtain plots of the ERA5 water vapor estimates. Each flight is discussed below:

– On January 14, the P-3 resampled a snow-band several times at different flight altitudes. VIPR measurements and ERA5 reanalysis fields display a repeating moisture band associated with the snow band. That is, both datasests show peaks of moisture of up to $5 \, \text{g m}^{-3}$ up to around 2 km altitude with widths of around 50 minutes (around 505 km assuming an average P-3 cruise speed of 607 km/hr). Qualitatively, VIPR and ERA5 are in good agreement between 2-6 km altitude.





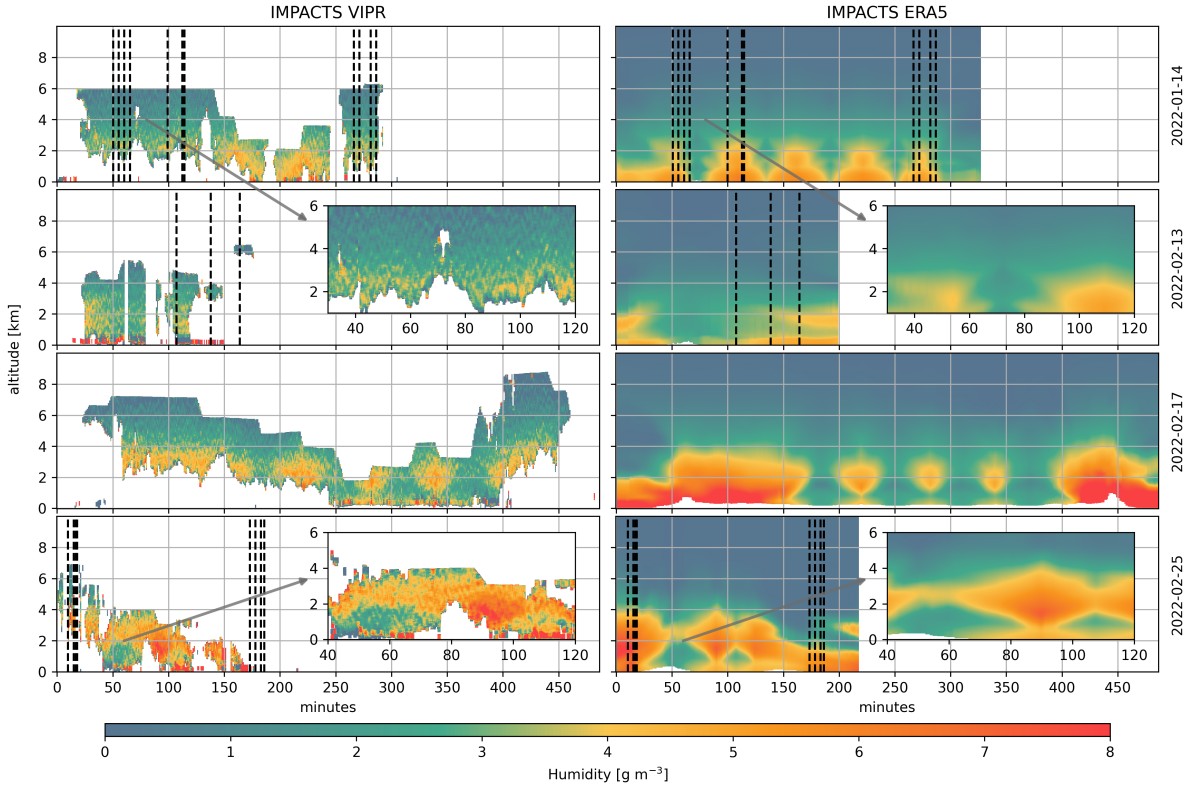

**Figure 7.** VIPR humidity measurements and ERA5 interpolated reanalysis fields for the IMPACTS flights shown in Figure 6-left. Dropsondes launches are indicated by dashed vertical lines. The insets showcase VIPR's ability to capture high-resolution humidity variations within the snowstorms. For these insets the raw VIPR data were smoothed using a 300 m running average in the vertical, as well as a 1 minute running average in the horizontal.

– On February 13, ERA5 fields display an intrusion of low moisture ($<3\,\mathrm{g\,m^{-3}}$) between 40 and 100 minutes into the flight
surrounded by higher moisture values ($4\text{-}5\,\mathrm{g\,m^{-3}}$). Although radar sampling is limited in this flight, VIPR measurements also hint at this dry layer. Note that, 120 minutes into the flight, VIPR measurements indicate clear sky conditions while the elevated humidity found in ERA5 may suggest some cloud formation.

– On February 17, VIPR and ERA5 datasets display moisture bands around 220, 280, 340 minutes into the flight, similar to the ones found on the January 14 flight. In the ERA5 interpolated fields two large high moisture regions (up to 8 g
$\mathrm{m^{-3}}$) can be seen at the beginning and end of the flights. VIPR was not able to sample these regions due to the significant attenuation associated with them but the upper section of these region is well represented by the VIPR measurements. As with the January 14 flight, VIPR and ERA5 agree about the location and magnitude of the elevated moisture band with values near $5\text{-}6\,\mathrm{g\,m^{-3}}$) at an altitude around 2 km.



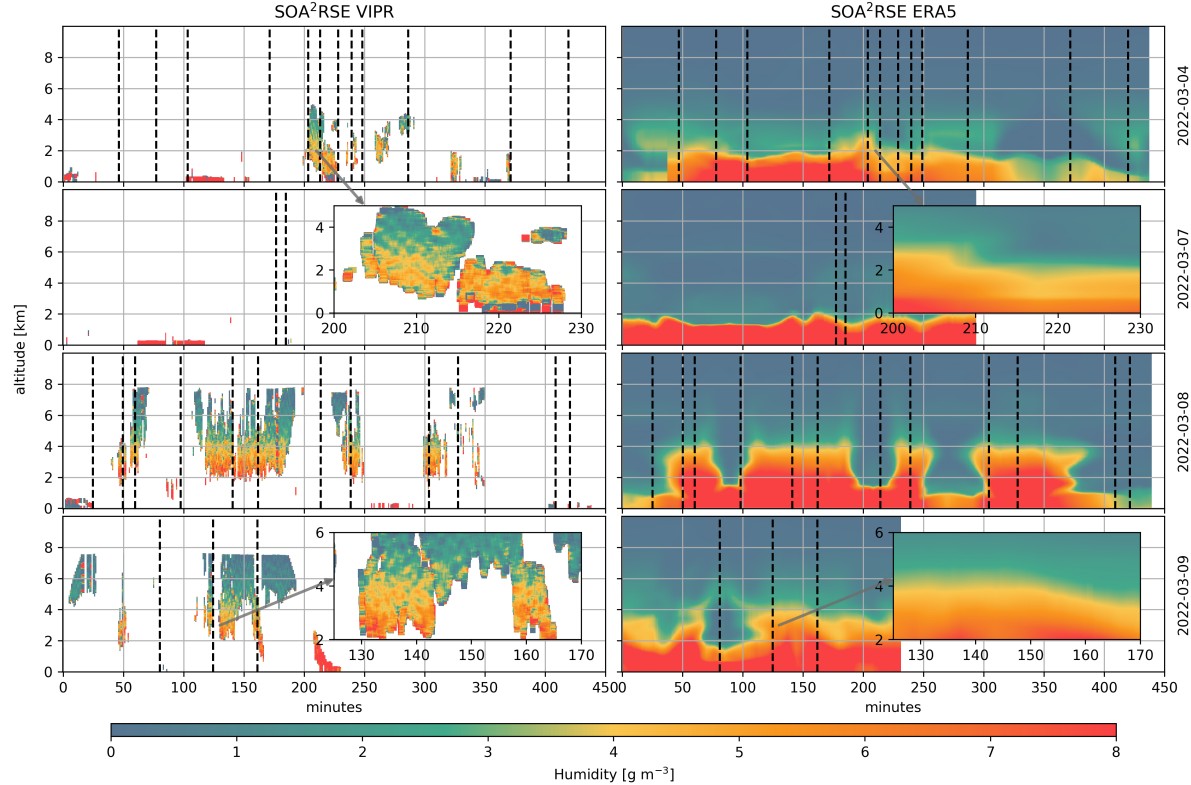

**Figure 8.** VIPR humidity measurements and ERA5 interpolated reanalysis fields for the SOA$^2$RSE flights shown in Figure 6-right. Dropsondes launches are indicated by dashed vertical lines. The insets showcase VIPR's ability to capture high-resolution humidity variations within the clouds. For these insets the raw VIPR data were smoothed using a 300 m running average in the vertical, as well as a 1 minute running average in the horizontal.

- On February 25, VIPR and ERA5 datasets display a region of high moisture with values up to 8 g m$^{-3}$ around 100 minutes into the flight. Note that, 160 minutes into the flight, VIPR measurements indicate clear sky conditions while the elevated humidity found in ERA5 may suggest some cloud formation.

- On March 4 (i.e., Figure 8) only a few isolated shallow congestus clouds were observed with VIPR. VIPR and ERA5 generally agree with regards to the humidity structure in the precipitating cloud near 200 minutes. However, VIPR suggests that ERA5 underestimates the water vapor in the cloud layer at 2 km altitude around 260 minutes. This is likely a decaying cumulus cloud with associated elevated water vapor relative to the background due to convective detrainment. In addition VIPR shows larger vapor values than ERA5 in the active cumulus around 320 minutes suggesting a slight dislocation of the ERA5 planetary boundary layer (PBL) horizontal moisture gradient.

- On March 7, the flight was entirely over clear sky so no VIPR profiles are available for a comparison.





- On March 8, VIPR and ERA5 measurements exhibit a strong humidity gradient at around 3 km throughout much of
the flight. The PBL is relatively moist with with values of absolute humidity near 8 g m$^{-3}$. The VIPR profiles are
     significantly attenuated beneath $\sim$2 km altitude and are in good qualitative agreement with ERA5 above this altitude.

- The March 9 flight is similar in character to the March 8 flight.

Figure 7 and 8 insets showcase VIPR's ability to capture high-resolution humidity variations within clouds and precipitation.
Although some of these variations may be due to detection noise and systematic biases, it is highly likely that the structured
variability is from real water vapor variations.

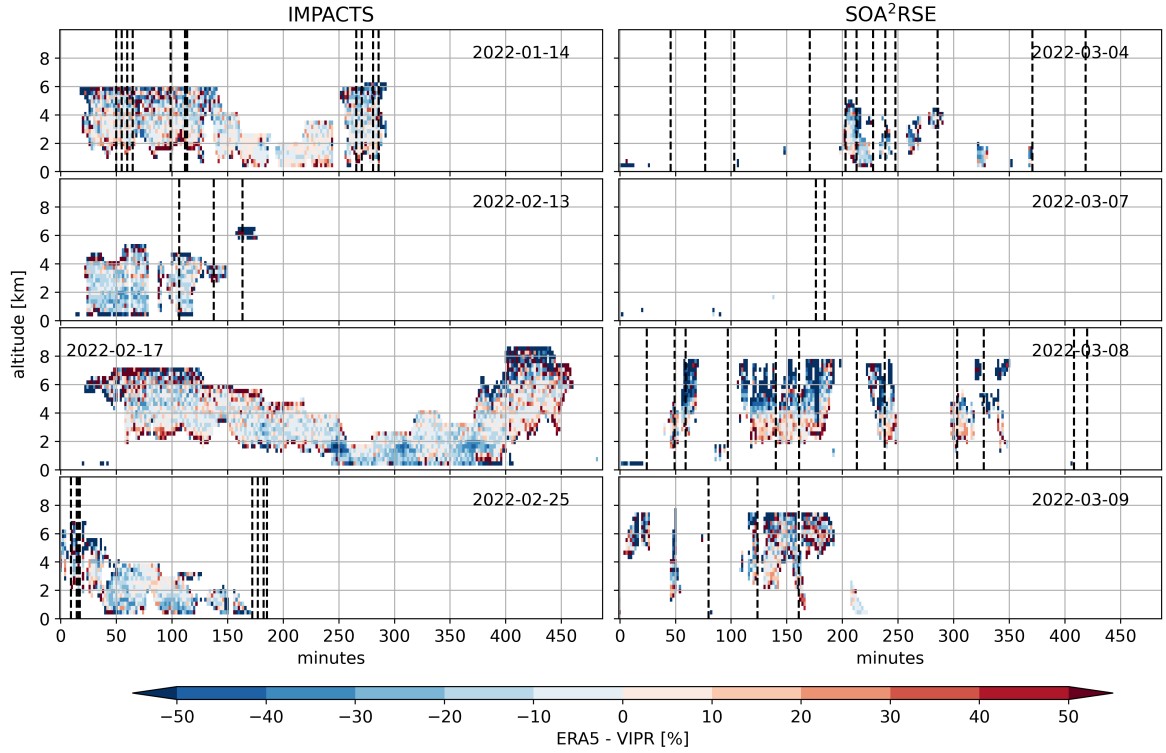

**Figure 9.** Percentage differences between the ERA5 and VIPR humidity estimates for the IMPACTS (right) and the SOA$^2$RSE (left) flights.

To round up the VIPR/ERA5 intercomparison, Figure 9 presents percentage differences between VIPR measurements and
ERA5 reanalysis fields. During IMPACTS, VIPR and ERA5 agrees within 20% except in the moisture bands in the February
17 flight, where ERA5 seems to be underestimating the VIPR estimates by up to 50%, as well as, at the cloud/snowtorm edges
(top and bottom) where VIPR retrieval artifacts could be the culprits of such differences. During SOA$^2$RSE, ERA5 display
up to a 50% underestimation of the VIPR estimates above $\sim$4 km. This underestimation is corroborated by the dropsonde
comparison as discussed below.



Figure 10 shows a comparisons with dropsondes for both VIPR measurements and ERA5 reanalysis fields to further explore their validity. This figure shows individual dropsondes through selected flights. VIPR measurements and ERA5 reanalysis fields are averaged for 10 minutes around the dropsondes release time, that is, 5 minutes before and 5 minutes after (allowing
to average up to 300 profiles, that is, approximately 100 km along track). This averaging, if random measurement errors are dominant over systematic errors (which is not generally the case), would in principle result in a water vapor retrieval precision as good as $\sim 0.07\,\mathrm{g\,m^{-3}}$ ($1.23/\sqrt{300}$).

Two VIPR estimates are shown, one using 3 radar tones that should alleviate the hydrometeor influence (Roy et al., 2021) and one using only two radar tones (167.12 and 174.74 GHz). The influence of hydrometeors is clearly discernible in the two-
tone estimates (green lines in Figure 10) for January 14 sondes 3 and 6, February 13 and 25 sonde 1, and all comparisons in March. In these cases, there is an overall high bias which can be attributed to frequency-dependent attenuation caused by hydrometeors being incorrectly attributed to water vapor. Note that the three-tone estimates (blue lines) effectively mitigate these biases and agrees much better with the dropsondes through all comparisons shown in figure 10. These results clearly indicate that the DAR retrievals should be considered most accurate when using the 3-frequency approach outlined in (Roy
et al., 2021) and demonstrated with data for the first time here.

Another distinct problem with both retrievals (using 2 or 3 radar tones) is a scattering effect most evident near cloud boundaries which can result in significant errors of either sign. This is particularly evident in February 13 sonde 1 and March 4 sonde 6. In these cases, even though we average over 10 minutes around the dropsonde, there were not enough valid retrievals to offset the artifacts.

Figure 11 displays scatterplots between VIPR, ERA5, and dropsondes color-coded by height. Note that we only compare the altitudes at which they were at least 150 VIPR water vapor estimates (less than half of the maximum number for the 10 minute window). The ERA5 comparisons are separated into cloudy and clear sky regions as determined by VIPR. Thus, the cloudy comparisons allow a direct intercomparison of ERA5 and VIPR against dropsondes. IMPACTS and SOA$^2$RSE flights are compared separately since they flew over completely different cloud regimes.

The VIPR comparisons (for either the IMPACTS or SOA$^2$RSE flights) when averaged over 10 minutes suggest a reasonably good agreement with the dropsondes (with a root mean square deviation, RMSD, better than $0.55\,\mathrm{g\,cm^{-3}}$, and an overall bias better than $0.05\,\mathrm{g\,cm^{-3}}$). Similarly, ERA5 agrees extremely well against the dropsondes during the IMPACTS-cloudy scenarios (RMSD=$0.13\,\mathrm{g\,cm^{-3}}$ and bias = $-0.08\,\mathrm{g\,cm^{-3}}$). However, the ERA5 does not agree nearly as well with the sondes during the SOA$^2$RSE-cloudy scenes (RMSD=$0.66\,\mathrm{g\,cm^{-3}}$ and bias =$-0.45\,\mathrm{g\,cm^{-3}}$), with ERA5 displaying clear underestimations for
values above 4 km. We speculate that the decreased fidelity of the ERA5 estimates in the SOA$^2$RSE campaign (in contrast with the IMPACTS ones) results from the fact that these flights took place in more subtropical latitudes characterized by isolated convection that is less well constrained by synoptic scale dynamics in the data assimilation system than the large scale mid-latitude snowstorms targeted during the IMPACTS flights. Specifically we speculate that the model's convective parameterization may not be producing enough convective transport of moisture out of the PBL into the lower free troposphere
during these flights.





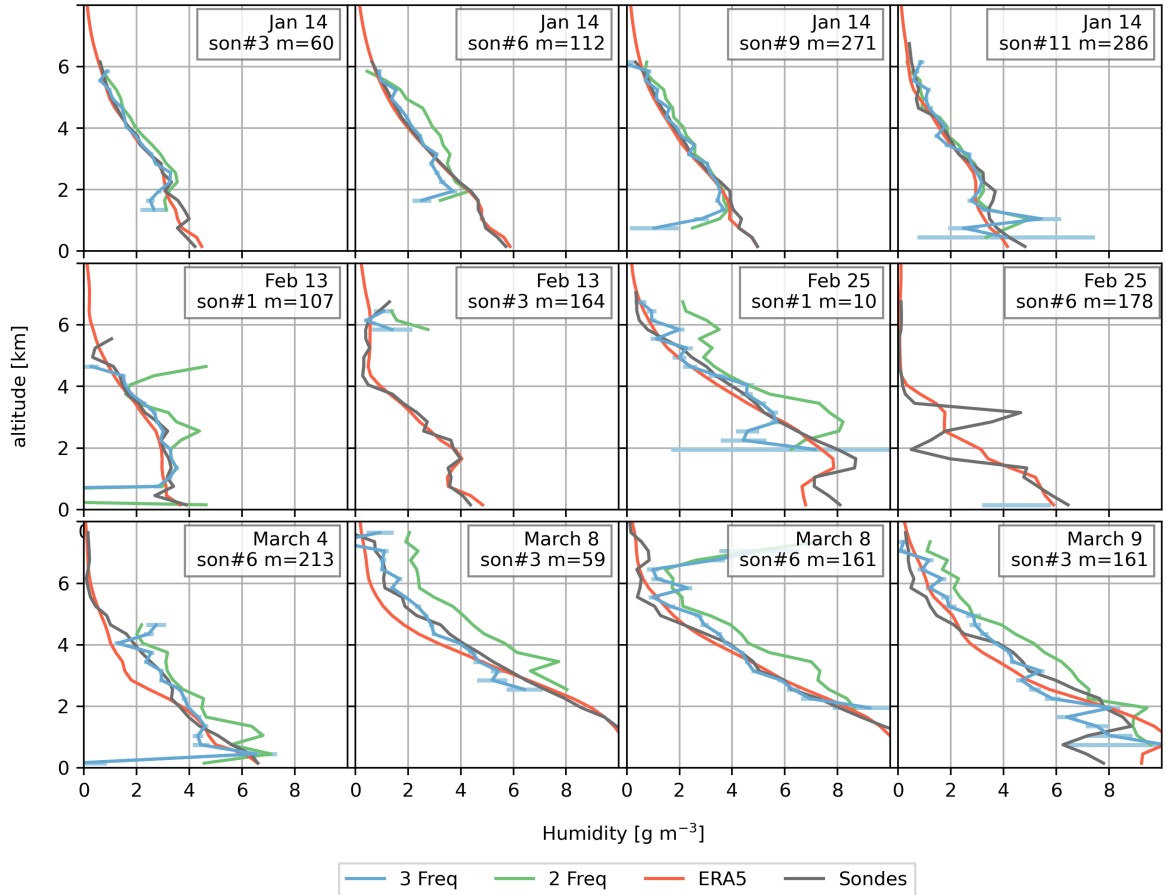

**Figure 10.** Profile comparisons between dropsondes, VIPR measurements, and ERA5 reanalysis fields. VIPR and ERA5 measurements were averaged over 10 minutes around the dropsonde time. Two VIPR estimates are shown, one using 3 radar tones to alleviate hydrometeor influence and one using only two radar tones (167.12 and 174.74 GHz). Error bars (i.e., the averaged random uncertainties) are shown for the 3 radar tones retrieval but they are barely visible due to the temporal averaging. Note that ERA5 and the dropsonde humidities were interpolated to the VIPR vertical grid. The legends display the flight date, sonde number, and the minutes into the flight when the sonde was dropped.

To explore this further, Figure 11-bottom summarizes the biases versus altitude between VIPR, ERA5, and the dropsondes. During IMPACTS, VIPR water vapor estimates agree within 10% with the dropsondes through most heights. ERA5 also agrees within 10% (or better) during the cloudy scenes. During SOA$^2$RSE, VIPR water vapor estimates agree within 20% with the dropsondes through most heights. However, there are some spikes around 6 and 7 km that can be mitigated by either using the median instead of the mean or by removing water vapor estimates that are more than 3 standard deviations from the mean when averaging over the 10-minute period used in these comparisons. ERA5 agrees within 20% with the dropsondes below





4 km. However, above ∼4 km ERA5 displays an underestimation of up to 50% in cloudy conditions, thus corroborating the VIPR comparison shown in Figure 9.

Under clear sky scenarios, ERA5 shows overall good agreement against the dropsondes (with RMSD better than $0.71 \, \mathrm{g \, cm^{-3}}$,

and an overall bias better than -0.04 $\mathrm{g \, cm^{-3}}$) but displays large excursions from the one to one line, as large as 350% in some instances. During IMPACTS, these excursions translate to biases of up to 250% around 2.5 km and up to ∼100% around 3.5 km.

Overall, the cloudy and clear-sky comparisons indicate good agreement between ERA5 and the dropsondes at the core of the snowstorms but not at the edges. That is, ERA5 is presumably failing to simulate the extent of the snowstorms likely due

to its temporal and horizontal resolution that, even-though they are the finest among the current reanalyses, they are still too coarse to resolve the rapidly changing snowstorm's extent. During SOA$^2$RSE, ERA5 agrees within 20% with the dropsondes below 5 km but shows an overestimation of up to 80%.





**Figure 11.** (top) Dropsonde humidity measurements scattered against VIPR and ERA5 estimates. The ERA5 comparisons are separated between cloudy and clear sky regions as measured by VIPR. (bottom) Average biases between VIPR, ERA5, and the dropsondes lauched during the IMPACTS (dark blue) and the SOA$^2$RSE flights (light blue). The dashed vertical gray lines indicates the ±20% difference while the dashed-dot vertical gray lines indicates the ±10% difference. Note that, VIPR and ERA5 humidities were average over 10 minutes around the dropsonde launch (approximately 100 km along track). All the available dropsondes were used on these comparisons, that is 22 and 29 dropsondes during IMPACTS and SOA$^2$RSE respectively.



## 5   Partial column results

Figure 12 shows the scatter between VIPR and dropsondes pCWV estimates for two sets of VIPR reflectivties, one using the
sphere calibration factors described in section 2.3 and the other employing a modified set of calibration factors (discussed
below). For this comparison, the VIPR measurements were temporally averaged by 10 minutes, 5 minutes before and after the
dropsonde launch. This amounts to averaging around 300 individual estimates, equivalent to $\sim$100 km (assuming an average
P-3 speed of 607 kmh$^{-1}$), with a theoretical error of around 0.03 g cm$^{-2}$ (i.e., 0.5 / $\sqrt{300}$). These comparisons encompass
clear-sky and cloudy scenes demonstrating the all-sky capability of DAR.

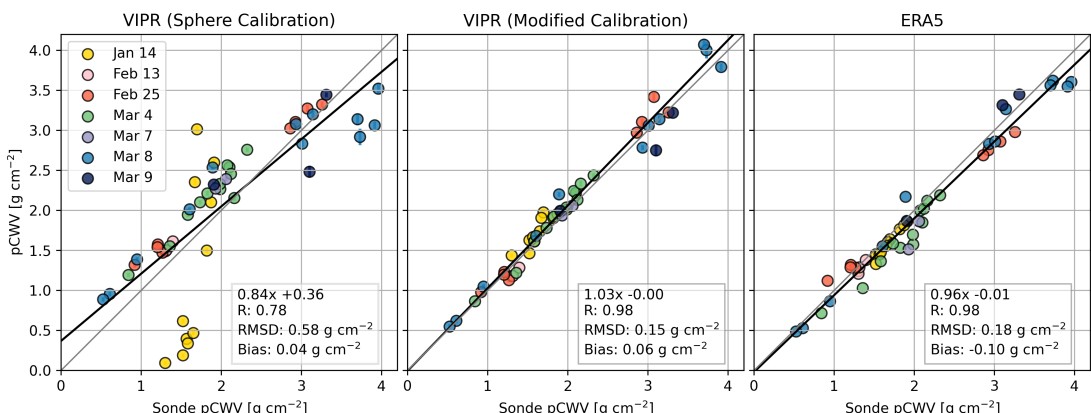

**Figure 12.** Dropsonde pCWV measurements scattered against the VIPR pCWV estimates using the sphere calibration (left) a modified
calibration (middle), as well as against the ERA5 pCWV. The gray line is the one-to-one line. The solid black line displays a linear fit. The
root mean square deviation, the linear fit equation, the bias are shown for each comparison. Measurements from different days are shown in
different colors. The VIPR errors are around 0.03 g cm$^{-2}$ (i.e., 0.5 / $\sqrt{300}$) which are smaller than the symbol size. There are 51 points on
these comparisons, one per each available dropsonde (22 during IMPACTS and 29 during SOA$^2$RSE).

A key difference between profiling with DAR and measuring the pCWV is that pCWV requires accurate relative calibration
of the radar frequencies. Figure 12-left shows the scatter between the dropsondes and the VIPR pCWV estimates that use the
sphere calibration factors as described in section 2.3. The best-fit line has a slope of 0.84, a RMSD of 0.58 g cm$^{-2}$, and a
correlation coefficient (R) of 0.78 with a bias of 0.04 g cm$^{-2}$. Although these results suggest a reasonable agreement between
the two datasets, they are biased by the estimates from the January 14 flight (yellow dots), which are the only estimates on this
figure using the calibration factor estimated on September 28, 2021 prior to the LNA failure. By simply changing the sphere
calibration factor of the 167.12 GHz tone from 8.9 to 12.9, the agreement against the dropsondes (see Figure 12-middle)
improved considerably. Similarly, by adjusting the sphere calibration factor (post LNA failure) of the 158.6 GHz tone from
10.7 to 12.2, the estimates for the rest of the flights aligned much better with the one-to-one line.





The modifications to the sphere calibration factors were found by trial an error. We emphasize that this recalibration pro-
cedure only ensures that the frequencies are calibrated among each other and should not be considered absolute calibrations.
Absolute calibration is not required for any DAR water vapor estimation but would be important in any effort to derive cloud
microphsycal properties. We are continuing work to investigate sources of ground-calibration uncertainty and improve its
reliability.

After applying the modified calibration factors (Figure 12-middle), there is a noticeable improvement in the comparison
metrics. The best-fit line slope increases to 1.03, and the RMSD decreases to $0.15\,\mathrm{g\,cm^{-2}}$, the correlation coefficient improves
to 0.98 while the bias becomes $0.06\,\mathrm{g\,cm^{-2}}$.

Note that the RMSD is much larger than the analytical VIPR uncertainty, $0.03\,\mathrm{g\,cm^{-2}}$, presumably due to a combination of
factors: (1) the sondes can be quickly advected away from the drop location (with horizontal displacements of up to $\sim$22 km
at the surface) which may introduce collocation errors. (2) During the 10 minute window, used for this comparison, even the
dropsondes disagree among each other due to the quickly changing nature of the snowstorms scenes. For example, the second,
third and fourth dropsondes during the January 14th flight, were dropped in a 10 minute span and have an RMSD (with respect
to their mean value) of $0.15\,\mathrm{g\,cm^{-2}}$; Similarly, the first four dropsondes and the last four dropsondes of the January 25th flight
were dropped in a 8 and 13 minute span, respectively, and have RMSDs of 0.15 and $0.13\,\mathrm{g\,cm^{-2}}$; which may explain the
RMSD values shown in the VIPR-sonde comparison (Figure 12-middle).

Figure 12-right shows the scatter versus the ERA5 pCWV estimates. These estimates are derived from the interpolated fields
to the VIPR times and locations. That is, they are averaged in the same fashion as the VIPR pCWV estimates to allow a direct
comparison. ERA5 shows an excellent agreement with the sondes measurements, with a best-fit line slope of 0.96, the RMSD
$0.18\,\mathrm{g\,cm^{-2}}$, the correlation coefficient 0.98, and a bias of $0.10\,\mathrm{g\,cm^{-2}}$.

To conclude the pCWV comparison, Figure 13 displays joint histograms of VIPR and ERA5 pCWV estimates (at the VIPR
native temporal resolution, i.e., 1.9 s). The regression analyses suggest either an underestimation by the ERA5 reanalysis or an
overestimation by the VIPR measurements for both IMPACTS and SOA$^2$RSE flights. Given that the VIPR measurements are
not expected to differ significantly from the measurements obtained around the dropsondes sites (as seen in Figure 12-middle),
we hypothesize that the slight disagreement is likely due to an ERA5 underestimation. We note that the VIPR precision error,
which is approximately $0.5\,\mathrm{g\,cm^{-2}}$ at this resolution, accounts for most of the RMSD. Further, other VIPR systematic effects
maybe at play, such as, drift in its sensitivity, or non-ideal surface scattering (e.g. non-random speckle averaging).

## 6 DAR and DIAL synergy

During the SOA$^2$RSE flights the High Altitude Lidar Observatory (HALO) water vapor DIAL (Carroll et al., 2022) flew
in conjunction with VIPR on the same P-3 aircraft to study the synergy of the DIAL and DAR techniques. HALO uses four
wavelengths spread among different strength absorption features in the 935 nm water vapor line complex to provide water vapor
sensitivity across the vastly different regimes in the troposphere and lower stratosphere. HALO also uses the high spectral
resolution and backscatter measurements at 532 nm and 1064 nm, respectively for cloud and aerosol profiling (Hair et al.,





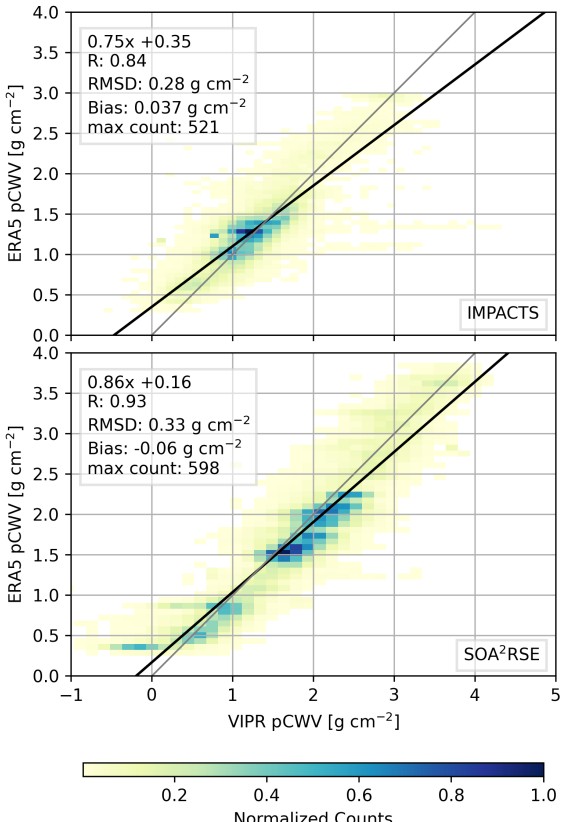

**Figure 13.** Two dimensional normalized histograms derived from VIPR and ERA5 pCWV during IMPACTS (top) and SOARSE flights (bottom). These histograms were computed at the native VIPR resolution (i.e., no averaging). The gray line is the one-to-one line. The solid black line displays a linear fit. The root mean square deviation, the linear fit equation, the bias, and the maximum number of counts are shown for each comparison.

2008; Carroll et al., 2022). The HALO water vapor retrievals presented here have a resolution of $300\,\mathrm{m} \times \sim 6\,\mathrm{km}$ vertical and horizontal resolution, respectively. The data are reported on a $15\,\mathrm{m}$ vertical grid and archive files subsampled from 0.5 seconds (2Hz) to every $10\,\mathrm{s}$ ($\sim 1.2\,\mathrm{km}$) to keep files sizes manageable. HALO water vapor profiles during SOA$^2$RSE exhibit random

uncertainties better than $0.125\,\mathrm{g\,m^{-3}}$ in regions of high moisture (i.e. $\geq 4\,\mathrm{g\,m^{-3}}$) and better than $0.05\,\mathrm{g\,m^{-3}}$ elsewhere.

Figures 14 showcases the synergy of the lidar and radar measurements as well as the DIAL and DAR measurements. The top panels display the high spectral resolution lidar aerosol backscatter at $532\,\mathrm{nm}$ for the the SOA$^2$RSE flight on March 8th and 9th while the middle panels show the VIPR radar reflectivities at $167.12\,\mathrm{GHz}$. As shown, as the clouds thicken the lidar becomes insensitive to largest particles while VIPR becomes sensitive to them. The bottom panels display curtain plots of the HALO

and VIPR water vapor estimates for the flights on March 8th and March 9th. The combined use of HALO and VIPR enables the estimate of high-resolution water vapor profiles in both clear sky and in-cloud conditions. This synergy could provide



insights into lower tropospheric clear sky and in-cloudy turbulent transport processes enabling comparisons and improvements of regional models as well as large eddy simulations.

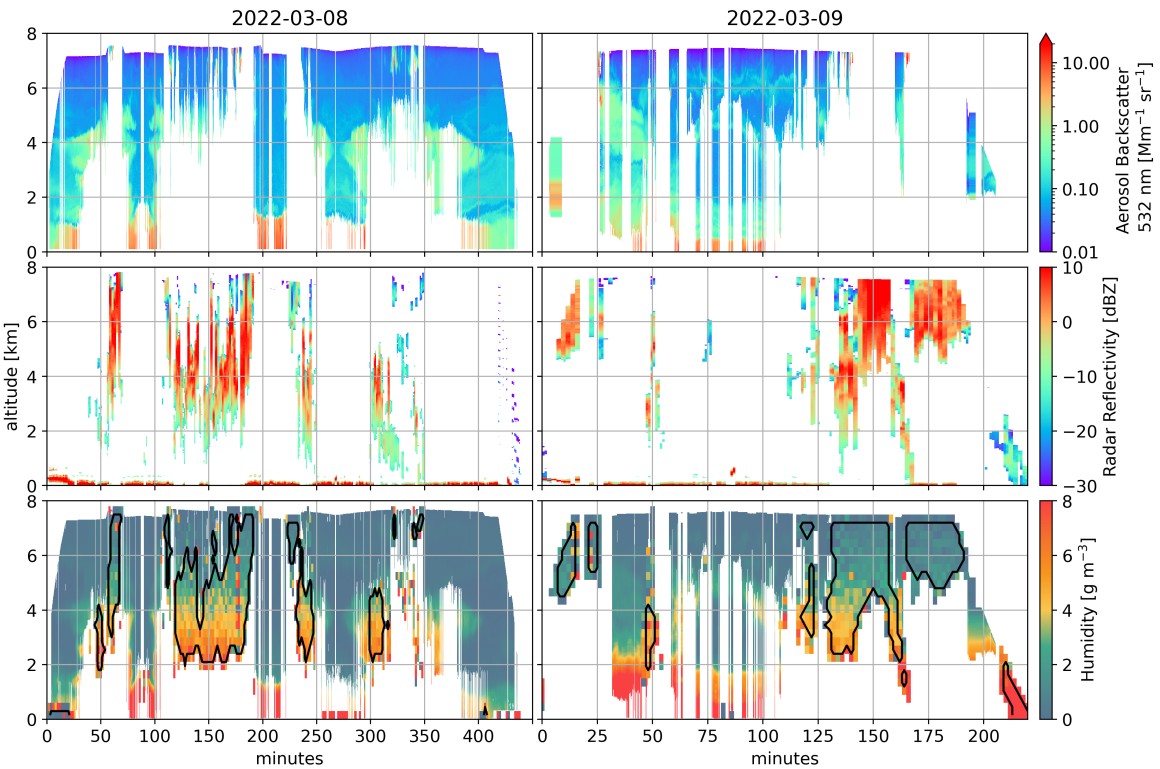

**Figure 14.** (top) 532 nm high-spectral-resolution lidar aerosol backscatter for the SOA$^2$RSE flight on March 8th and March 9th, (middle) calibrated radar reflectivities at 167.12 GHz (also shown in Figure 6), and (bottom) VIPR and HALO humidity measurements. VIPR measurements are delimited by a black contour.

## 7 Conclusions

The VIPR G-band radar was deployed on the P-3 aircraft as part of the IMPACTS and SOA$^2$RSE campaigns, encompassing a total of 8 flights and collecting approximately 44 hours of data. This provided the opportunity of evaluating the VIPR DAR water vapor estimates under two complete different weather regimes, that is, inside snowstorms (IMPACTS flights) and during calm weather (SOA$^2$RSE flights).

  For the first time, VIPR operated using three radar tones (158.6, 167.12, and 174.48) which allowed to disentangle the
differential extinction from the water vapor from the hydrometeor scattering and absorption effects. By using three frequencies



instead of only two (as in previous VIPR configurations), biases were mitigated, ensuring more accurate and reliable water vapor estimations.

In this study, VIPR and ERA5 averaged to 10 minutes along track were intercompared and validated against dropsondes. Thess comparisons can be broken in three categories:

— *In-cloud profile comparisons:* During IMPACTS, VIPR and ERA5 agrees within 20% except at the moisture bands where ERA5 seems to be underestimating the VIPR humidty measurements by up to 50%. During SOA$^2$RSE, ERA5 display up to a 50% underestimation of the VIPR estimates above ∼4 km.

A scatterplot comparison against dropsondes suggest an overall good agreement with VIPR during both IMPACTS and SOA$^2$RSE (with RMSD better than 0.55 g cm$^{-3}$ and an overall bias better than 0.05 g cm$^{-3}$) and an overall
good agreement with ERA5 during IMPACTS (RMSD=0.13 g cm$^{-3}$ and bias = -0.08 g cm$^{-3}$). During SOA$^2$RSE, ERA5 shows a clear underestimation for water vapor (RMSD=0.66 g cm$^{-3}$ and bias = -0.45 g cm$^{-3}$), in particular for estimates above ∼4 km. Breaking down this into biases versus altitude suggest that, during IMPACTS, VIPR humidty estimates agree within 10% with the dropsondes through most heights while ERA5 agrees within 10% (or better). During SOA$^2$RSE VIPR water vapor estimates agree within 20% with the dropsondes through most heights, while ERA5 agrees
within 20% with the dropsondes below 4 km, but displays an underestimation of up to 50% above it, thus corroborating the VIPR and ERA5 intercomparison.

Through these comparinsons, it was evident that the VIPR retrieval may induce some artifacts at the clouds/snowtorm edges. Biases induced by these artifacts may be mitigated by either using the median instead of the mean or by removing water vapor estimates that are more than 3 standard deviations from the mean when averaging over the 10-minute period
used in these comparisons.

— *Clear-sky (ERA5 only) profile comparisons:* During both IMPACTS and SOA$^2$RSE, ERA5 shows overall good agreement against the dropsondes (with RMSD better than 0.71 g cm$^{-3}$, and an overall bias better than -0.04 g cm$^{-3}$) but displays large excursions from the one to one line. These excursions translate to large biases (up to 250%) at particular heights. Since ERA5 agrees with the dropsondes in the in-cloud regimes, this clear-sky biases suggest that ERA5 is pre-
sumably failing to simulate the extent of the snowstorms). During SOA$^2$RSE, ERA5 agrees with the dropsondes within 20% below 5 km but displays an overestimation of up to 80% above this altitude.

— *pCWV comparisons:* Overall, VIPR and ERA5 display a good agreement with the dropsondes pCWV estimates, with VIPR displaying a RMSD of 0.15 g cm$^{-2}$ and a bias of 0.06 g cm$^{-2}$ while ERA5 displays a RMSD of 0.18 g cm$^{-2}$ and a bias of -0.1 g cm$^{-2}$. These relatively high RMSD values are presumably due to the nature of the snowstorm scene, that
is, dropsondes launched within ten minutes of each other, also display an RMSD of around 0.15 g cm$^{-2}$ with respect to their mean value. A direct comparison between the VIPR and ERA5 pCWV estimates indicate a possible ERA5 underestimation, in particular during the IMPACTS flights.



Lastly, the SOA$^2$RSE campaign offered, for the first time, the opportunity to show the synergistics abilities of combining DIAL and DAR water vapor measurements enabling the estimate of high-resolution water vapor profiles in both clear sky and
in-cloud conditions.

*Author contributions.* LM perform the analyses and wrote the manuscript. KC, RD, JS, and RRM developed the VIPR instrument. KC and RD estimated the calibration factors and installed VIPR on the P-3. KC developed the noise subtraction techniques. ML, JS, and LM collected the VIPR data. AN was the PI for SOA$^2$RSE. RBG and ARN collected the HALO data. ARN and JC processed the HALO data. CR, KT, and HV launched the dropsondes. All authors commented on the manuscript.

*Competing interests.* The authors declare that they have no conflict of interest.

*Acknowledgements.* This research was carried out at the Jet Propulsion Laboratory, California Institute of Technology, under a contract with the National Aeronautics and Space Administration (80NM0018D0004). Support for the SOA$^2$RSE mission was provided by the NASA Earth Science Technology Office and the NASA Earth Science Division Weather and Atmospheric Dynamics focus area. The authors wish to thank the P-3 aircraft and ground support crew for making the IMPACTS and SOA$^2$RSE flights a success. The HALO authors acknowledge
the contribution of Brian Carroll in serving as flight scientist and preliminary processing of HALO data for the SOA$^2$RSE mission.



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
