# Peer review of "Water Vapor Measurements inside clouds and storms using a Differential Absorption Radar"

_EGUsphere, 2023_

## Author Comment (AC1)

We thank the reviewers for their comments. Below are our responses in blue.

Here is a summary of the major changes in the revised manuscript:

- In the introduction we added a discussion on previous water vapor measurements.
- We modified Figure 1 and Figure 3 to minimize the number of panels.
- Panels from Figure 4 and Figure 5 were reshuffle as requested by reviewer 2.
- The flight-by-flight descriptions of the radar reflectivities and the water vapor curtain were deleted.
- ERA5 is now compared in the model resolution.
- We added the melting layer to Figures 6,7,8,9, and 14.
- The units on the partial column results were changed from $g\ cm^{-2}$ to $kg\ m^{-2}$
- We added a figure showing the Normalized histogram of the humidity retrievals during the March 8th and March 9th flights using either DAR/VIPR or DIAL/HALO to further highlight the synergy of the DAR and DIAL techniques
- As an appendix figure we added the March 4 DIAL/DAR synergy.

**Response to Reviewer 1**

The comments of "egusphere-2023-1807 Water Vapor Measurements inside clouds and storms using a Differential Absorption Radar" by Millán et al.

This article mainly uses the Differential absorption radar to measure the water vapor content in clouds and storms, the authors analyze the VIPR humidity measurements during two NASA field campaigns: (1) the Investigation of Microphysics and Prsecipitation for Atlantic Coast-Threatening Snowstorms (IMPACTS) campaign, with the objective of studying wintertime snowstorms focusing on East Coast cyclones; and (2) the Synergies Of Active optical and Active microwave Remote Sensing Experiment (SOA2RSE) campaign which studied the synergy between DAR (VIPR) and differential absorption lidar (DIAL, HALO) measurements. The comparison with the reanalysis data is also discussed. The results of this paper are undoubtedly of great significance for the measurement of the water vapor content in the cloud. The paper language expression is also good. Nevertheless, there are still some issues that need to be revised or clarified. Specific comments are as follows:

(1) The definition of differential absorption technology in the abstract can be considered into the introduction or section 2, because the differential absorption technology is relatively familiar to most professional readers of atmospheric measurement technology, and the quantitative research conclusions can be added in the abstract to clarify the scientific results of this work.

DAR is a relatively new technique. While it may be familiar to some readers of AMT, we believe the casual reader may not be familiar with this technique. Thus, we decided to leave the brief DAR description in the abstract.

We added to following to the abstract:   Overall, in-cloud and in-storm comparisons suggest that ERA5 and VIPR agree within 20% or better against the dropsondes. The exception is during SOA2RSE (i.e., in fair weather), where ERA5 exhibits up to a 50% underestimation above 4 km.

(2) In the first paragraph of the introduction, the discussion on the progress of water vapor measurement is lacking. It is suggested to increase the new technical progress and existing problems in this aspect, and the reference of response should be added.

We added: Radiosondes provide the longest record but have limited spatial and temporal coverage, with only a few locations and launches per day (e.g., Wang et al., 2000). In-situ aircraft measurements are restricted to flight level (e.g., Zahn et al., 2014; Singer et al., 2022), while aircraft remote sensing options are limited to a few field campaigns (e.g., Johansson et al., 2018). Passive microwave or near-infrared spaceborne methods have been valuable in providing global information (e.g., Andersson et al., 2007). However, all spaceborne techniques have limitations: imagers only provide integrated column water vapor, lacking vertical distribution information, while sounders have broad weighting functions near the Earth's surface, limiting their vertical resolution. Infrared (or near-infrared) techniques are limited to clear-sky scenes, thus restricting coverage in the tropics. Radio-occultation techniques can provide high vertical resolution water vapor profiles, but their measurement geometry results in an averaging over more than 100 km horizontally. Furthermore, atmospheric ducting effects associated with the top of the boundary layer limit their accuracy in this region (e.g., Ao et al., 2012).

(3) The second paragraph of the introduction on the scientific objectives of these two projects (NASA two field campaigns) and the issues to be addressed in this paper need to be strengthened.

Section 2 of the paper "VIPR and the IMPACTS and SOAR2SE campaigns" provide these details.

(4) Table 1, The technical parameters required to increase the response such as signal to noise ratio, lowest detection line, detection distance and detection sensitivity, and suggest add a physical physical picture VIPR system and Hardware composition diagram.

We added in table 1 the noise-equivalent reflectivity at 1km range. As mentioned in section 2.2 (the Radar detection confidence flag) we only consider returns with SNR > 3.

We added after the ground based VIPR deployment explanation: (A simplified block diagram of this iteration of VIPR can be found in Cooper et al. (2021))

We added after the discussion of the two identical reflectors as separate primary apertures: A picture of VIPR mounted on the bomb-bay of the P-3 can be found at Cooper (2022).

(5) Figure 1 What is the basis of setting the flight trajectory?

In the caption we added:  Flight tracks of the P-3 VIPR measurements during the 2022 IMPACTS deployment (solid lines), **flight trajectories were selected to intersect snowstorm systems**. (bottom) Flight tracks of the P-3 VIPR and HALO measurements during the SOA$^2$RSE deployment (dashed lines), **flight trajectories were aimed towards a range of clear sky and cloudy conditions over the Western Atlantic Ocean.**

(6) Figure 3. Shows the Power spectrum examples at 167.12 GHz for a clear sky and a cloudy scene, Do the other two frequencies (158.6, 174.74 GHz) have a similar conclusion and use the same data processing method?  Yes, we added the following in the caption: The power spectra at 158.6 and 174.7 GHz are similar, with the former being affected by less water attenuation and the latter by more.

(7) Where the Equation 3 comes from?  We added Cooper et al., 2022 in the discussion of this equation.

(8) In section 3 Retrieval methodology and datasets used for comparisons, recommended to add a flow chart.

Reviewer 2 suggested many changes to reduce the article length, hence we decided not to include a flowchart.

(9) section 4 Vapor Profile Results, personal feeling it is a bit like an experimental report, rather than a scientific research paper, it is suggested to increase the regularity of the conclusion or the discovery of the elaboration, to improve the academic nature of the paper.

To avoid sounding like an experimental report, the flight-by-flight description of the radar reflectivities was deleted. The number of sondes per flights was added to table 2.

We also deleted the flight-by-flight description of the water vapor curtains. Instead we added the following text: Overall, VIPR and ERA5 are in good qualitative agreement. Both datasets depict moisture bands (i.e., high moisture regions) associated with snow bands on the January 14, February 17, and February 25 flights; a dry layer (moisture <3 g m−3) between 40 and 100 minutes into the February 13 flight; and a strong humidity gradient at around 3-4 km throughout much of the March 8 and 9 flights.  All the conclusions are discussed in the main manuscript.

---

## Author Comment (AC2)

We thank the reviewers for their comments. Below are our responses in blue.

Here is a summary of the major changes in the revised manuscript:

- In the introduction we added a discussion on previous water vapor measurements.
- We modified Figure 1 and Figure 3 to minimize the number of panels.
- Panels from Figure 4 and Figure 5 were reshuffle as requested by reviewer 2.
- The flight-by-flight descriptions of the radar reflectivities and the water vapor curtain were deleted.
- ERA5 is now compared in the model resolution.
- We added the melting layer to Figures 6,7,8,9, and 14.
- The units on the partial column results were changed from g cm$^{-2}$ to kg m$^{-2}$
- We added a figure showing the Normalized histogram of the humidity retrievals during the March 8th and March 9th flights using either DAR/VIPR or DIAL/HALO to further highlight the synergy of the DAR and DIAL techniques
- As an appendix figure we added the March 4 DIAL/DAR synergy.

**Response to Reviewer 2**

This study presents airborne measurements collected by the VIPR instrument during two recent field studies, IMPACTS and SOA2RSE. The authors describe the retrieval to derive water vapor profiles and partial column estimates by using the Differential Absorption Radar technique, accounting for differential hydrometeor scattering by including a third frequency. Water Vapor estimates are compared to dropsondes and ERA5 fields. The complementing nature of DAR and DIAL is illustrated in examples from the SOA2RSE flights.

**General Comments**

Overall, I find this paper to be a nice contribution to the newly emerging DAR G-band radar field. It is impressive to see how well the VIPR system performs from an airborne platform. The language of the paper is clear, especially the methodological part is well written, and Figures are generally clear. Yet, I think that the scientific content of the paper can be enhanced by sharpening the results sections. The analysis of the DAR-DIAL synergy in particular, a major novelty in the field, deserves a more thorough quantification and analysis.

**Specific Comments**

- LL 1-28: the introduction should contain more references to available literature, and a more thorough introduction on the VIPR instrument, as well as available literature on ERA-5 evaluation.

As requested by Reviewer 1 we added: Radiosondes provide the longest record but have limited spatial and temporal coverage, with only a few locations and launches per day (e.g., Wang et al., 2000). In-situ aircraft measurements are restricted to flight level (e.g., Zahn et al., 2014; Singer et al., 2022), while aircraft remote sensing options are limited to a few field campaigns (e.g., Johansson et al., 2018). Passive microwave or near-infrared spaceborne methods have been valuable in providing global information (e.g., Andersson et al., 2007). However, all spaceborne techniques have limitations: imagers only provide integrated column water vapor, lacking vertical distribution information, while sounders have broad weighting functions near the Earth's surface, limiting their vertical resolution. Infrared (or near-infrared) techniques are limited to clear-sky scenes, thus restricting coverage in the tropics. Radio-occultation techniques can provide high vertical resolution water vapor profiles, but their measurement geometry results in an averaging over more than 100 km horizontally. Furthermore, atmospheric ducting effects associated with the top of the boundary layer limit their accuracy in this region (e.g., Ao et al., 2012).

We modified the following sentences:

- Active water vapor sounding techniques such as differential absorption lidar (DIAL) **(e.g., Browell et al., 1983; Wulfmeyer and Bösenberg, 1998; Behrendt et al., 2009; Carroll et al., 2022)** and differential absorption radar (DAR) **(e.g., Lebsock et al., 2015; Millán et al., 2016; Cooper et al., 2018; Roy et al., 2018; Battaglia and Kollias, 2019)**, have been proposed as potential solutions …

- Previous studies have demonstrated the efficacy of DIAL and DAR in estimating water vapor profiles **from an aircraft platform** (e.g., Nehrir et al., 2017; Roy et al., 2022).

- In this study, we present an analysis of airborne water vapor estimates obtained with the Vapor In-Cloud Profiling Radar (VIPR) **(e.g., Cooper et al., 2018; Roy et al., 2022)** during two field campaigns.

- We validate VIPR measurements of water vapor against collocated dropsonde measurements and the **ERA5** reanalysis fields **[Hersbach et al., 2020)].**

VIPR and ERA5 are discussed in sections 2 and section 3.5 respectively. Thus, we added the following:  This paper is organized as follows: Section 2 describes the VIPR measurements and aircraft campaigns, Section 3 discusses the retrieval methodology and datasets used in the comparisons, Section 4 and 5 cover the profiling and partial column results, and Section 6 describes the DAR and DIAL synergy.

- L 152: are there thin clouds that the radar is insensitive to, or which are filtered out by the phase noise model? Characterizing VIPR's sensitivity in more detail could also be highlighted in the synergy with DIAL (see comments Sec. 6).

The short answer is yes, there are thin clouds to which VIPR is not sensitive. This limitation is now precisely documented in Table 1 which now shows the noise-equivalent reflectivity. Furthermore, the new Figure 15 shows the complementarity of DIAL and DAR in terms of absolute humidity by showing the distributions of retrieved values from the two instruments.

Also, we change the sentence:

This criteria allow us to discriminate spurious returns that at first sight could be consider clouds when in reality is just phase noise rising above the noise floor.

To: While we might lose details about thin clouds, these criteria are designed to effectively filter out any spurious returns caused by phase noise rising above the noise floor.

Sensitivity information should be added to Tab 1.

We added in table 1 the noise-equivalent reflectivity at 1km range.

- Sec. 3.1: The 2-frequency DAR method is described in detail, summarizing previous literature (Roy et al 2018, Roy et al 2020, Battaglia and Kollias 2019). The analyses in the paper highlight the benefit of using the 3-frequency DAR method to mitigate differential scattering effects on the water vapor retrieval. I would recommend to shorten the 2-frequency description as this method has been described in detail in literature, while the modifications necessary to the retrieval to embed the 3rd frequency should be highlighted, e.g. in Eq (13) and LL205-210.

Both methods are discussed in the literature. We chose to explain the 2-frequency retrieval as it is conceptually easier to understand. Readers interested in exploring the methodology further are directed to the relevant literature. We believe that with the details given the reader can understand that when using more frequencies the retrieval can *partly disentangle the differential extinction from the water vapor from the hydrometeor scattering and absorption effects as shown by Battaglia and Kollias (2019) and Roy et al. (2022).*

We added the following sentence at the beginning of section 3.1:  The DAR retrieval methodology is fully discussed elsewhere (Roy et al., 2018; Battaglia and Kollias, 2019; Roy et al., 2021, 2022). For completeness, here we provide a recap of a simplified retrieval to provide a heuristic understanding of the minimal DAR physics. The DAR profiling technique begins by …

- L 252: The authors should clarify why they don't use the results with improved precision in their analysis instead.

We modified the text: We acknowledge that the chirp-down estimates are nearly identical but display some signs of distortion, the cause of which is under investigation.

Further, considering the averaging times (and the corresponding relatively small precision values) and the magnitude of the systematic biases there is no point in using the chirp up / chirp down average. We added the following sentence to the text: However, considering the magnitude of the possible systematic biases and differences between the datasets discussed in Section 4, such averaging is deemed unnecessary.

- L 281: I understand that ERA5 gives an hourly snapshot with a resolution of 31km; airborne data is averaged to 10km and 1min. How do the authors account for differences due to these temporal and spatial scales of the VIPR-ERA5 comparison? By interpolating the model to the airborne data, the model output gets oversampled. In order to avoid such spatial and vertical oversampling of the model pixels, the comparison should rather be performed on model resolution. This would additionally allow to quantify the model's sub-pixel variability. How strongly does the performance of VIPR-ERA5 depend on this variability?

All comparisons are now made on the model resolution.

We deleted: To ease comparison, the ERA5 humidity fields were interpolated to the VIPR measurement times and locations. And added: To ease comparison, we identified the nearest ERA5 humidity field spatially and temporally.

We also modified the captions deleting all mention to the interpolation of ERA5.

- I recommend to edit the figures in Secs 2 and 3 to increase clarity and reduce the manuscript's length. More specifically, I would suggest to:

   - combine Fig 1a and b in one panel
       Thank you for this suggestion, Figure 1 was changed as requested.

[Figure]

- drop Fig 2: readily available in literature (eg Roy et al 2018, Fig 3). While we agree that this figure is available elsewhere, we believe is important to have it so the reader understand the chirp up/chirp down difference. Many readers of this paper will likely not be experts in FM radar techniques which are essential to understand Fig 3.

- Fig 3: add bottom row to upper panel to reduce to two panels Again, thank you for this suggestion, Figure 3 was changed as requested.

[Figure]

In the caption we added: The labels highlight features in the chirp up power spectrum, while mirror features can be found in the chirp down power spectrum.

- reshuffle Figure 4 and 5, e.g. by adding the bottom panel of Fig 5 below Fig 4; and combining most right panel Fig 4 with three panels Fig 5.

The figures were reshuffled as requested:

[Figure]

With the caption now reading: Uncalibrated VIPR 167.12 GHz reflectivities measured during the first ~40 minutes of the January 14 flight with no noise subtraction (top), with noise subtraction (middle), and with noise subtraction and confidence flag (bottom). The noise subtraction is indicated by the pink/green bar (for phase noise and thermal noise, respectively).

[Figure]

With the caption now reading: (left) Radar reflectivities at the three slices indicated in Figure 4, depicting scenarios with clear sky (S1, navy line, featuring a bright reflection around 6.5 km), a partly cloudy scene (S2, light blue), and a cloudy scene (S3, green). (right) Phase noise model simulations for each of these slices. The integrated phase noise model is depicted in dark gray, while phase noise model examples at different ranges are shown in light gray.

- Section 4:  To tighten the argument, I would suggest to re-order the analysis presented in this Section as follows: first, evaluate the VIPR retrieval with dropsondes, highlighting differences between 2- and 3-frequency retrieval; then, assess ERA-5 with airborne measurements.

After careful consideration, we have decided to keep the layout as it is. The current arrangement enables a seamless transition from the radar reflectivity curtains (Figure 6) to the water vapor curtains (Figures 7 and 8). The suggested re-order, as recommended by the reviewer, might lead to a less fluid reading experience, jumping from radar reflectivities to sondes and back again. Importantly, it's worth noting that the suggested re-order does not alter the manuscript's content.

- LL295 – 312: I propose to move some of this information to the description of the flights in Tab 2, LL73, respectively, to focus on the scientific messages.

We deleted those lines. We modified table 2 to include the number of sondes per flight. We also included the number of sondes under cloudy and precipitating conditions as measured by VIPR.

We deleted the sentence: For example, note the attenuation or disappearance of the surface return below regions of heavy hydrometers burden, such as on the February 17 flight around 420 minutes into the flight.

And added: For example, noticeable attenuation is observed throughout much of the flight on January 14 below 2 km due to precipitation. Similar attenuation is observed below regions with a heavy hydrometeor burden, as seen during the February 17 flight, especially around 420 minutes into the flight.

- LL313 – 348 and Fig 7 and 8: The analysis presented here, in my view, remains very descriptive. My suggestion is to pick a couple of case studies in which main caveats of the VIPR-ERA5 comparison are highlighted, even in a timeline comparison. I would suggest to plot VIPR at top, ERA5 at bottom on the same time axis to ease the visual intercomparison. A time line of VIPR's water vapor could instead be added to Fig 6 to give the reader a direct illustration of reflectivity and water vapor product.

We deleted the flight by flight description.

We added: Overall, VIPR and ERA5 are in good qualitative agreement. Both datasets depict moisture bands (i.e., high moisture regions) associated with snow bands on the January 14, February 17, and February 25 flights; a dry layer (moisture <3 g m−3) between 40 and 100

minutes into the February 13 flight; and a strong humidity gradient at around 3-4 km throughout much of the March 8 and 9 flights.

We also changed the inset discussion to: Figure 7 and 8 insets demonstrate VIPR's ability to capture high-resolution humidity variations within clouds and precipitation. While some of these variations may be due systematic biases, the presence of structured variability strongly suggests real water vapor variations.

It would be great to have some additional measurement overview statistics summarized in a Table, e.g.: how many dropsondes measured in cloudy, precipitating or clear conditions?

Table 2 now includes the number of sondes per flight as well as the dropsondes under cloudy/precipitating conditions.

How many VIPR columns were affected by attenuation part-way through the column?

The answer to this depends on the type of scene measured by VIP: under heavy precipitation, most of them will be affected, while under fair weather, most of them will not. To provide a representative answer, we would need to simulate the current VIPR configuration for global conditions, which is outside the scope of this article.

- Figs 6-9: adding the height of the melting layer would be an important additional piece of information.

We added the melting layer to the figures as requested (see updated manuscript). We also added the melting layer to figure 14 (the DAR/DIAL curtains).

We also added the following text to the caption:  Magenta dashed lines depict the melting layer derived by interpolating the ERA5 reanalysis fields to the VIPR measurement times and locations.

- LL 349: the authors state the detection noise as one of the reasons for mis-matches between VIPR and ERA-5 water vapor. The lidar measurements should be used to quantify the detection noise; and to assess the impact on the comparison to ERA5.

We deleted *detection noise* because we meant precision noise, and the precision noise is relatively small at this time resolution. The sentence now reads: . While some of these variations may be due systematic biases, the presence of structured variability strongly suggests real water vapor variations.

- LL 358: The authors should motivate the chosen 10 minute averaging interval. How sensitive is the comparison to this interval? I think that the evaluation of the VIPR retrievals with respect to dropsondes should be analysed before the ERA5 assessment (see comment above) as retrieval evaluation.

Below are the corresponding figure 10 and 11 showing the comparisons averaging 2 minutes (i.e., one minute before and one minute after). As shown the results are similar but noisier especially for the SOA2RSE differences in the figure 11 equivalent (first panel, bottom row of the second figure below).

Note that these comparisons uses all the available points while the 10 minutes averages (shown in the paper) only uses bins with at least 150 VIPR water vapor estimates (less than half of the maximum number for the 10 minute window) to ensure a cloudy volume.

[Figure]

[Figure]

In the manuscript we added: The 10-minute average was chosen to reduce noise, especially when comparing it against the radiosondes dropped during SOA2RSE, where the clouds were scattered, in contrast to the continuous cloud cover during IMPACTS.

- Fig 10: Please motivate the choice of dropsondes presented here. I would suggest to highlight attenuation effects as well as differential scattering impacts by illustrating the retrieval performance for examples of profiles with attenuation effects; with best retrieval performance; and for solid/liquid precipitation occurrence.

Dropsosondes were selected to showcase a variety of retrieval performances, in particular to showcase the different magnitude of the biases between the 2 and 3 frequency retrievals. We also aimed to showcase at least one retrieval for each flight (with cloudy measurements).

In the manuscript we deleted: This figure shows individual dropsondes through selected flights.

And added: Dropsondes were selected to demonstrate varying retrieval performances, particularly highlighting biases between the 2 and 3 frequency retrievals (see below). We included at least one dropsonde from each flight with cloudy measurements.

A study of retrieval performance over solid or liquid precipitation occurrence is currently outside the scope of this paper.

- LL 371: I wonder if one way around this could be to sample the dropsondes depending on the conditions at launch (cloudy, clear, precipitating) to avoid artefacts. Please also state how many dropsondes were used for the presented statistical comparison (see eg comment LL313 with suggestion to include a table).

Since VIPR only measure under cloudy/precipitating conditions it is impossible to avoid that problem. The number of sondes under cloudy / precipitating conditions are now shown in table 2.

- Section 6 is very short and vague compared to Sec 4 and 5 while it is in my opinion one of the most novel findings of the paper. The authors should thoroughly quantify the synergistic benefits of the two instruments. The following questions could guide the analyses: How well do clear-sky pCWV agree between the two instruments? What happens at the edges from clear-sky to cloudy, when DIAL and DAR water vapor profiles follow one another temporally? How well do water vapor observations agree above cloud top (DIAL profile vs DAR pCWV)? I would suggest to add a Figure similar to Fig 13 to analyse the clear-sky pCWV.

A thorough quantification of the synergistic benefits of DIAR and DAR are outside the scope of this manuscript. That said, we added the following figure to further highlight the synergy of the two techniques.

[Figure]

Figure 15: Normalized histogram of the humidity retrievals during the SOA$^2$RSE March 8th and March 9th flights using either DAR/VIPR or DIAL/HALO. The numbers in brackets represent the number of samples per instrument.

We also added: The combined use of HALO and VIPR enables the estimate of high-resolution water vapor profiles in both clear sky and in-cloud conditions. For example, on the March 8th flight, HALO can see in between the clouds (see around 90, 200, and 275 minutes into the flight), revealing elevated water vapor values (~8g m$^{-3}$) in the PBL (the first 1 km of the atmosphere) while next to it, VIPR indicates such elevated values up to around 3 km within the mid-level convection (See also Figure 8, March 8th ERA5 panel).

To underscore the synergy of these techniques, Figure 15 displays the normalized histograms of the water vapor values retrieved by either DIAL/HALO or DAR/VIPR. As shown, DIAL/HALO displays a skewed right distribution with mode at 0.4 gm$^{-3}$ (attributed to a majority of clear sky measurements),  while DAR/VIPR exhibits a multimodal broad distribution with modes at 0.8 and 4.4 gm$^{-3}$.  This synergy could …

- Fig 14: It is nice to get an overview of the different measurements and retrieved water vapor profiles, but it is hard to catch details of the instrument synergy from the current Figure. A Figure should be added, e.g. to zoom in on the DIAL and DAR-derived water vapor curtains around cloud edges to illustrate the different vertical resolutions and quality of the retrieved profiles.

As mentioned above, a thorough quantification of the synergistic benefits of DIAR and DAR are outside the scope of this manuscript.

The authors should also comment on the performance of the synergy for the 03-04 and 03-07 flights. We added the figure below as an appendix figure.

[Figure]

In the manuscript we added: Similarly, Figure A1 displays the March 4th flight, note that there were no cloudy measurements on the March 7$^{th}$ flight.

- The authors should add a paragraph in the Conclusions on future perspectives available from the presented measurements and results.

The last sentences of the conclusion now read: Observations of water vapor in the planetary boundary layer from space are crucial for advancing our understanding of the dynamics in this critical atmospheric layer (National Academies of Sciences and Medicine, 2018). These airborne campaigns constitute an essential step toward transitioning DAR and DIAL to an orbital platform.

**Technical Corrections**

- SI units should be used for water vapor and pCWV throughout the entire manuscript (kgm$^{-2}$; gm$^{-3}$).

Now we are using kg m-2 and gm-3.   Figure 12 and 13 were updated.

- L 154: clarify where the reader should be directed to in the figure. We changed "(after minute 37)"  to "(after minute 37, bottom right corner) in the 2 km closest to the surface."

- L 200 ff: sentence not complete.  The following was deleted "In principle, all 3 with a weak frequency dependence."   The information was conveyed in the previous sentence.

- L 231: should read: if more than […].  Corrected

- LL 380 and further, also conclusion: see above; unit should be gm$^{-3.}$  Thank you for finding that typo, the typo has been corrected and now all those units read gm$^{-3}$

- L407: „shows and overestimation of up to 80%" relative to what?  The sentence now reads: During SOA$^2$RSE, ERA5 agrees within 20% with the dropsondes below 5 km but shows an overestimation of up to 80% **at higher altitudes**.

- L420: introduce LNA abbreviation.  We added (LNA) in the last sentence of section 2.3. "The re-calibration was done because part way through the campaign, VIPR's low-noise amplifier **(LNA)** failed and was replaced."

- Fig 11 caption contains multiple typos. Corrected

- L479: „Thess" should be „these".   Corrected

- L492: contains multiple typos.   Corrected